# Ultrathin covalent organic overlayers on metal nanocrystals for highly selective plasmonic photocatalysis

Anubhab Acharya [1,2], Trimbak Baliram Mete[1,2], Nitee Kumari[1,2], Youngkwan Yoon[2], Hayoung Jeong[3], Taehyung Jang[4], Byeongju Song[3], Hee Cheul Choi[2], Jeong Woo Han[3], Yoonsoo Pang [4], Yongju Yun[3], Amit Kumar [1,2] ✉ & In Su Lee [1,2,5] ✉

Metal nanoparticle-organic interfaces are common but remain elusive for controlling reactions due to the complex interactions of randomly formed ligand-layers. This paper presents an approach for enhancing the selectivity of catalytic reactions by constructing a *skin-like* few-nanometre ultrathin crystalline porous covalent organic overlayer on a plasmonic nanoparticle surface. This organic overlayer features a highly ordered layout of pore openings that facilitates molecule entry without any surface poisoning effects and simultaneously endows favourable electronic effects to control molecular adsorption–desorption. Conformal organic overlayers are synthesised through the plasmonic oxidative activation and intermolecular covalent crosslinking of molecular units. We develop a light-operated multicomponent interfaced plasmonic catalytic platform comprising Pd-modified gold nanoparticles inside hollow silica to achieve the highly efficient and selective semihydrogenation of alkynes. This approach demonstrates a way to control molecular adsorption behaviours on metal surfaces, breaking the linear scaling relationship and simultaneously enhancing activity and selectivity.

Controlling metal nanostructures' sizes, shapes, crystalline phases, multi-elemental compositions and metal–metal/support/ligand interfaces can endow tunable optoelectronics and high catalytic performance[1–6]. However, it is still challenging to regulate the molecular adsorption behaviours on the metal surface, breaking the linear scaling relationship, and simultaneously enhancing the activity and selectivity[7,8]. Unlike homogeneous catalysts and enzymes where the nature of ligand–metal interaction determines the reaction selectivity, the predesigned role of organic surface modifiers on heterogeneous catalyst surfaces has limited applicability[9–11]. Interfacial steric, electrostatic and electronic effects of capping ligands and self-assembled

monolayers on nanoparticles (NPs) can influence different reaction selectivities[7–10]. However, random conformational crowding by the densely bound ligand chains can also result in catalytic surface poisoning and poor conductivity of electrons/holes[12–15]. Additionally, loosely bound ligands on the metal surface undergo difficult-to-control adsorption–desorption behaviours in response to reactants, under thermal or light-irradiation conditions[16–20]. Ligand-free NPs are prone to deform and aggregate, thus losing their nanocharacteristic optical and catalytic properties. Confining NPs inside the rigid porous shells of silica or metal/covalent-organic frameworks (MOFs/COFs) endows selectivity, mainly because of the specific features of shell

[1]Creative Research Initiative Center for Nanospace-confined Chemical Reactions (NCCR), Pohang University of Science and Technology (POSTECH), Pohang 37673, Korea. [2]Department of Chemistry, Pohang University of Science and Technology (POSTECH), Pohang 37673, Korea. [3]Department of Chemical Engineering, Pohang University of Science and Technology (POSTECH), Pohang 37673, Korea. [4]Department of Chemistry, Gwangju Institute of Science and Technology (GIST), Gwangju 61005, Korea. [5]Institute for Convergence Research and Education in Advanced Technology (I-CREATE), Yonsei University, Seoul 03722, Korea. ✉e-mail: amitkumar@postech.ac.kr; insulee97@postech.ac.kr

components in controlling substrate orientation, size-based molecular sieving and shell-specific noncovalent interactions[21–34]. However, owing to the huge difference in the structural dimensions of NPs, thick shells and trapped ligands/polymers between them, the nature of the two-dimensional interface remains difficult to control and poorly utilised. Moreover, thick bulk-sized (usually micron-scale) porous shells tend to impede molecular diffusion-dependent reaction rates and render a narrow substrate size scope.

We aimed to maximise the interfacial synergy between the metal surface and the organic modifiers by constructing a *few-nanometre* ultrathin orderly porous covalent organic overlayer (pCOL) directly on the ligand-free plasmonic nanocube (NC) surface. In the desired favourable features of pCOL, (i) a highly ordered layout of well-defined pore openings would facilitate consistent molecular transport without any complicated surface poisoning effect and (ii) conformationally restricted crosslinked organic units would intimately align with the surface metal atoms to optimise the steric and electronic effects. We utilised an isolated ligand-free plasmonic (Ag/Au) NC inside porous hollow silica ($h$-SiO$_2$) for on-surface light-induced oxidative activation and successive covalent crosslinking assembly to form a conformal few-molecule-thick overlayer (<5 nm), while avoiding the formation of a typical microsized thick shell. Previously, we modified ligand-free plasmonic NC surfaces with a conformal catalytic metal atomic layer by light-induced metal-growth chemistry[35]. Plasmonic-catalytic hybrids possessing localised surface plasmon resonance (LSPR) can induce various challenging reactions through sustainable and energy-efficient utilisation of charge carriers, but achieving control over reaction selectivities is difficult[17,36–38]. In a distinct advancement, the present study addresses the challenge of controlling the reaction selectivity in plasmon-induced catalysis. By utilising pCOL interfaced Pd/AuNC inside $h$-SiO$_2$, we performed light-induced semihydrogenation of various substituted alkynes to Z-alkenes (up to >99% yield) with high selectivity (up to >99%). The selective semihydrogenation of alkynes to

alkenes is a fundamentally challenging and industrially significant synthetic transformation[39–42]. Our strategy overcame the commonly observed overhydrogenation at extended reaction times while achieving complete conversions and exhibited nearly consistent performance with various mono and disubstituted alkynes. In-depth studies based on a series of control experiments, in situ surface-enhanced Raman scattering (SERS) and density functional theory (DFT)-based calculations revealed the mechanistic details of the favourable optimum electronic and steric microenvironment of pCOL in controlling the reaction selectivity.

## Results and discussion

### Synthesis of pCOL on plasmonic-catalytic NP surface

For the intended metal NP surface modification, we selected COF layers that can be easily synthesised through aldehyde-amine 'condensation-crosslinking-assembly' [Fig. 1, details in Supplementary Information (SI)]. Our preliminary attempts to use the as-synthesised surfactant-capped silver NCs (AgNCs) directly resulted in uncontrollably thick and poorly aligned COF shells (Supplementary Fig. 1). This is in agreement with the previously studied interference of embedded capping ligands in COF/MOF shells[30,31]. An alternative attempt involving COF shell formation on AgNCs resulted in severe aggregation, shape deformation and loss of LSPR properties (Supplementary Fig. 2). This led us to consider our previously developed method to access individual surfactant-free NCs inside the $h$-SiO$_2$, which are easily employable in the reactions without aggregation and lead to the selective deposition of the photochemical metal atomic layer on the NC surfaces[35]. We envisioned that a suitable parallel strategy involving photoinduced COF crosslinking can be initiated at the plasmonic NC surface, while ensuring that the precursors do not react off-surface, and the thickness of the COF layer can be controlled to a few nanometres by the limited availability of the plasmonic surface (Fig. 2a). Accordingly, we obtained ligand-free single isolated AgNCs

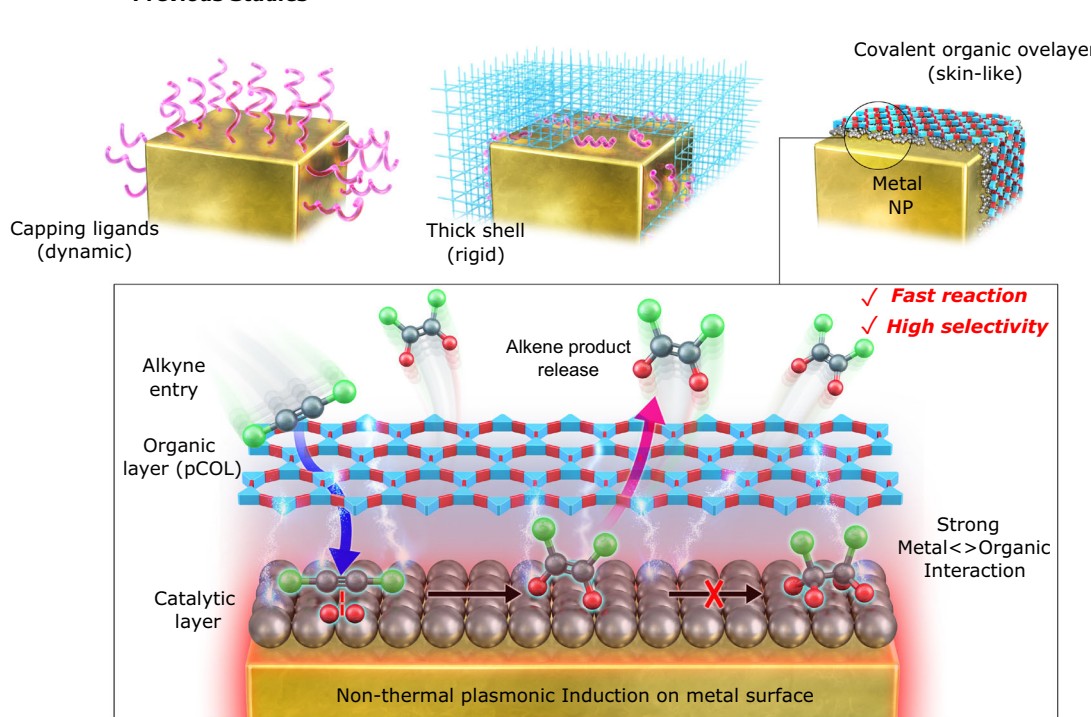

**Fig. 1 | Graphical representation of previously known different metal-organic interfaces and present design of covalent organic overlayer (pCOL)-metal interface controlling the selectivity of alkyne semihydrogenation.** Flexible capping ligands and rigid thick shells uncontrollably occupy the metal surface. In comparison, the present design of intimately interfaced ultrathin pCOL endows facile molecular transport and strong interfacial electronic interaction to afford fast reaction and high selectivity in plasmonically-induced catalysis.

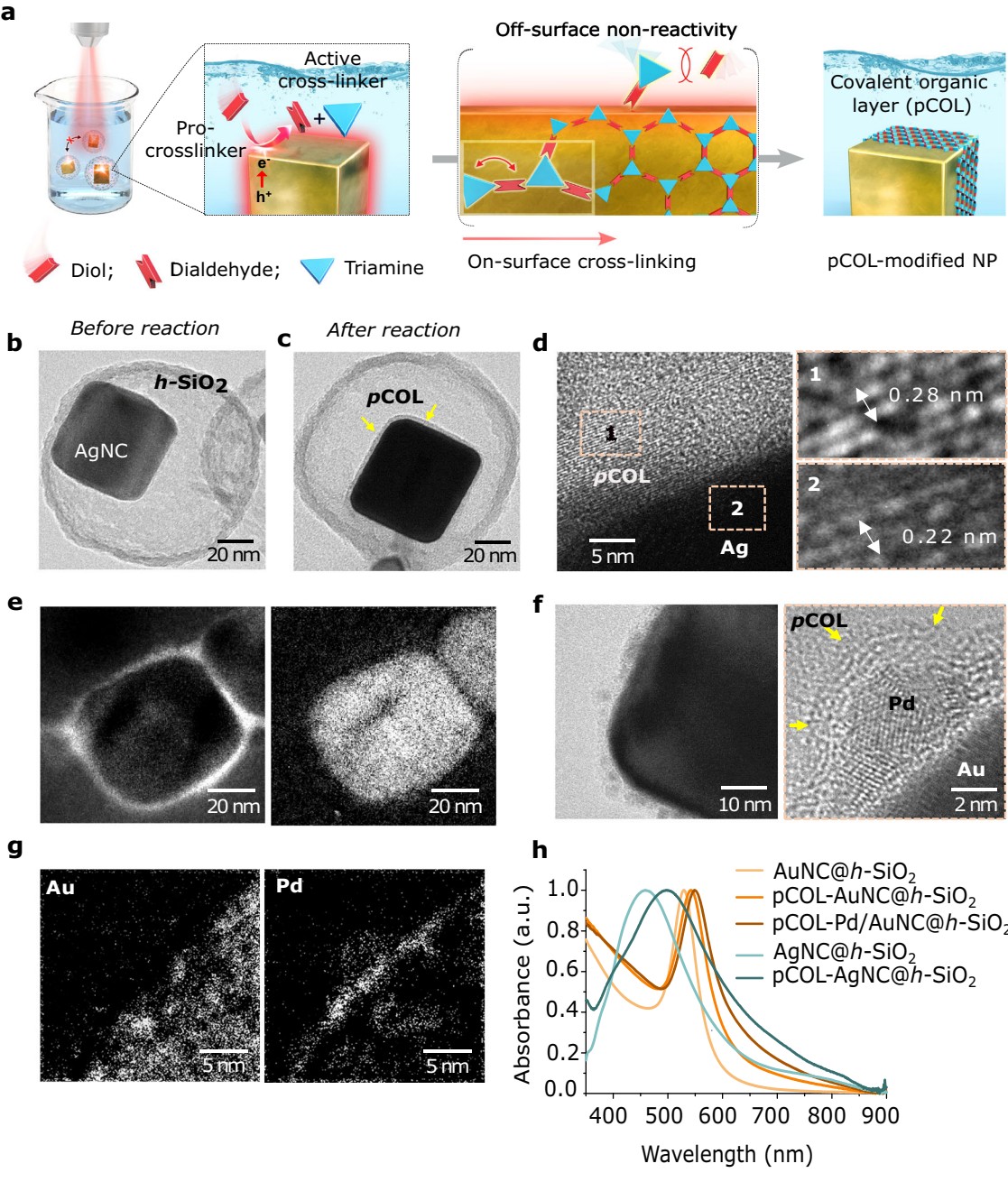

**Fig. 2 | Light-assisted pCOL modification on plasmonic NP surface. a** Schematic for the synthetic strategy. **b**–**d** TEM and HRTEM images of AgNC@*h*-SiO₂ before and after pCOL modification at different magnifications (mag): high-mag HRTEM images showing well-aligned AgNC-pCOL interface. **e** EELS elemental mapping of pCOL-modified AgNCs (SiO₂-free). **f** HRTEM images and **g** EELS elemental mapping of AuNCs after pCOL and Pd modification, high-mag HRTEM image showing curved pCOL over Pd. **h** UV–vis spectra of different plasmonic NCs before and after pCOL modification.

(edge size $50 \pm 2$ nm) inside *h*-SiO₂ (designated as AgNC@*h*-SiO₂), which were *ca.* 100 nm in size (Supplementary Fig. 3). A porous, thin and hollow silica shell (*ca.* 100 nm size, 7 nm thickness) ensured the colloidal stability of the AgNCs to avoid any aggregation-induced changes in the LSPR properties, as confirmed by NP tracking analysis (NTA), transmission electron microscopy (TEM) and ultraviolet–visible (UV–vis) spectrophotometry (Supplementary Fig. 4). In a typical procedure for modifying pCOL on AgNCs, we immersed AgNC@*h*-SiO₂ in a solution (dioxane: mesitylene = 4:1) containing diol (DAL, 1,4-benzenedimethanol) and triamine [TAE, 1,3,5-tris(4-aminophenyl)benzene] and exposed it to a blue laser (405 nm, 0.3 W/cm²) for 1 h. After reaction, the UV–vis spectrum exhibited broadening and a redshift ($\Delta\lambda = 40$ nm), indicating the change in the dielectric environment on

the AgNC surface through pCOL modification (Fig. 2). High-resolution TEM (HRTEM) and high-angle annular dark-field scanning TEM (HAADF-STEM) visualised the smooth and conformal organic polymeric layer (thickness $2.4 \pm 0.5$ nm) perfectly aligning with the AgNC surface (Fig. 2). Additionally, X-ray photoelectron spectroscopy (XPS) and electron energy loss spectroscopy (EELS)-based elemental mapping confirmed the presence of a nitrogen-containing organic overlayer on the surface (Fig. 2 and Supplementary Fig. 5). The presence of characteristic C = N peaks in the Raman and Fourier-transform infrared (FTIR) spectra at 1570 and 1647 cm⁻¹, respectively, verified the amine-aldehyde condensation (Supplementary Fig. 6). The difference in the Raman and FTIR frequencies may be due to complex parameters, including charge transfer on the metal surface. In a separate reaction,

[1]H NMR confirmed the AgNC-mediated oxidation of DAL to dialdehyde (DAE, 1,4-benzenedicarboxyaldehyde) upon laser irradiation (Supplementary Fig. 7). Direct condensation of DAE and TAE resulted in the formation of a bulk COF around AgNC@$h$-SiO$_2$ (Supplementary Fig. 8). Additionally, replacing light by heat (100 °C) in the mixture of DAL and TAE resulted in isolation of polymeric particulates (*ca.* 10 nm) from the AgNCs (Supplementary Fig. 9). The use of a near-infra-red laser (808 nm) or deoxygenated solutions containing DAL and TAE did not result in any reaction with AgNC@$h$-SiO$_2$ (Supplementary Fig. 10). Based on these results, we propose the following mechanism of pCOL formation: in response to blue laser irradiation, Ag-induced oxidation of DAL to DAE follows an *on-surface* covalent crosslinking with TAE to form extended porous aromatic rings containing flat sheets assembled to form *few-nanometre* ultrathin covalent organic layers; successive accumulation of organic layers on the NC surface slowed the rate of DAL oxidation and self-limited the thickness of pCOL without any excess growth at extended times (Supplementary Fig. 11). In a separate experiment, we confirmed the photochemical oxidative role of the Ag surface in the polymerisation of methyl methacrylate (MMA), resulting in a poly-MMA shell around the AgNC (Supplementary Fig. 12). The use of plasmonic NCs of different shapes (spherical, pyramidal and dodecahedral), metals (Ag, Au), or molecular compositions of the COF had minimal influence on the conformal COF overlayering process (Supplementary Fig. 13). We propose that pCOL formation follows a photochemical pathway instead of a thermal pathway. In our experiments, we used a 405 nm laser to induce the formation of pCOL, which is applicable for both AgNC (LSPR resonant) and AuNC (interband transition); however, a 532 nm laser (LSPR resonant) could also be applied in the case of AuNC, achieving similar results (Supplementary Fig. 14).

Furthermore, for catalytic applications, we synthesised a ternary hybrid interfacial system, including a catalytic metal (Pd)-modified plasmonic surface (AuNC) covered with pCOL inside $h$-SiO$_2$ (Fig. 2). In this designed catalytic nanoreactor (pCOL-Pd/AuNC@$h$-SiO$_2$), the effect of pCOL on reaction selectivity can be reliably studied: plasmonic AuNC will function as a nanoantennae, supplying highly localised energy flow to induce the reaction specifically at the interfacial [Pd]-sites and $h$-SiO$_2$ will ensure aggregation-free homogeneous dispersion of the catalyst[17,35–38]. For this, we extended our previously developed metal lamination strategy on pCOL-modified AuNC inside $h$-SiO$_2$ (designated as pCOL-AuNC@$h$-SiO$_2$) by immersion in ethylene glycol (EG) containing Na$_2$PdCl$_4$ under laser irradiation (532 nm, 0.3 W/cm$^2$)[35]. Highly localised plasmonic photochemical reduction of Pd(II) to Pd(0) by EG selectively modified only limited amounts of Pd as a thin layer (<2 nm) directly on the AuNC surface under pCOL as confirmed by HRTEM, HAADF-STEM, EELS and EDS elemental mapping (Fig. 2, Supplementary Figs. 15 and 16). Notably, the pCOL layer simply rearranged its location on the Pd-modified AuNC top surface similar to a flexible tight jacket and remained undisturbed because of the extended crosslinked organic network (Fig. 2). Despite the crystalline nature of the ultrathin pCOL being clearly characterised by the HRTEM images, the corresponding X-ray diffraction (XRD) peaks were below the detection limit. However, we verified the characteristic XRD signals and Brunauer–Emmett–Teller (BET) surface area analysis data for thicker pCOL synthesised on an exposed AuNP surface (Supplementary Figs. 17–19).

## Catalytic reactions studies for selectivity control

Next, in a typical catalytic hydrogenation reaction, we added pCOL-Pd/AuNC@$h$-SiO$_2$ to a solution (methanol) of diphenylacetylene (**1a**) and NH$_3$·BH$_3$ as the hydrogen source and exposed the reaction to a laser (405 nm, 0.3 W/cm$^2$). [1]H NMR and GC–MS based reaction kinetics studies monitored >99% conversion of alkyne **1a** to Z-alkene (**1b**) (*ca.* 99% yield, TOF = 63 min$^{-1}$) within 60 min (Fig. 3a, b and Supplementary Figs. 20–23). Interestingly, after complete conversion, extending the

reaction time (up to 100 min) resulted in only a small amount of alkane (**1c**) (<1%). In comparison, Pd NCs (without Au) modified with pCOL (Supplementary Figs. 24 and 25) and commercial Pd-on-carbon (Pd/C) at 60 °C exhibited early stage (at 100% **1a** conversion) over-hydrogenation of **1b** to **1c** up to >50% and >99%, respectively, and eventually converted to over-reduced alkane **1c** (>99%) (Fig. 3c, d and Supplementary Fig. 25). To study the role of each component in pCOL-Pd/AuNC@$h$-SiO$_2$, we performed test reactions with different control catalysts: Pd-AuNCs without pCOL inside $h$-SiO$_2$ (Pd/AuNC@$h$-SiO$_2$), AuNCs inside $h$-SiO$_2$ (AuNC@$h$-SiO$_2$), pCOL-modified AuNCs inside $h$-SiO$_2$ (pCOL-AuNC@$h$-SiO$_2$) and TAE-modified Pd/AuNCs inside $h$-SiO$_2$ (TAE-Pd/AuNC@$h$-SiO$_2$) (Fig. 3c and Supplementary Fig. 15). The absence of pCOL in Pd/AuNC@$h$-SiO$_2$ resulted in a similar reaction rate (TOF 60 min$^{-1}$) as pCOL-Pd/AuNC@$h$-SiO$_2$; however, partial over-hydrogenation (**1b**: **1c** = 60: 40), indicated the crucial role of pCOL in stopping overhydrogenation as a side reaction (Fig. 3c, d and Supplementary Fig. 26). AuNC@$h$-SiO$_2$ and pCOL-AuNC@$h$-SiO$_2$ did not result in any measurable reactions, verifying the indispensable role of Pd in catalytic hydrogenation and the nonreactivity of Au and pCOL. Upon replacing laser irradiation with dark thermal conditions at different temperatures, pCOL-Pd/AuNC@$h$-SiO$_2$ exhibited variable reaction rates and **1b** selectivities (Supplementary Fig. 27). For instance, increasing the temperature to 60 °C afforded high conversions (>99%), similar to the laser-induced reaction, but resulted in the loss of **1b** selectivity (<70%). In addition, continuing the low-temperature reaction for longer periods (>24 h) resulted in poor **1b** selectivity (Supplementary Fig. 28). In a separate study under laser irradiation, increasing the laser flux had a much lower adverse effect on **1b** selectivity, whereas high conversions were observed even at low laser fluxes (Supplementary Fig. 29). However, raising the bulk solution temperature under light irradiation caused a significant loss of **1b** selectivity (Supplementary Fig. 30).

Distinctly, under laser irradiation conditions, high conversions in conjunction with high alkene product selectivity strongly suggest a favourable contribution of nonthermal effects have a major contribution. Through Raman thermometry analysis (details in SI), we estimated the exact temperature on the plasmonic catalyst's surface not to rise more than 50 °C under moderate continuous wave laser fluxes (details in SI, Supplementary Fig. 31). These data again indicated that plasmonic nonthermal effects played a dominant role in catalytic induction, instead of the contribution from the photothermal pathway. The presence of pCOL protected **1b** from overhydrogenation effectively only in the case of plasmonically induced reactions, whereas thermal conditions promoted overhydrogenation regardless of the presence of pCOL. Adverse effects of temperature on semihydrogenation are already known, and they seem to dominate in the case of dark conditions[43]. As previously studied, LSPR-induced highly localised energy flow can selectively activate interfacial catalytic sites through the coupling of s electrons in the plasmonic metal to d electrons in the transition metal, resulting in the direct localisation/excitation of charge carriers in the transition metal shell without raising reaction temperatures[17,35–38,44,45]. Switching 'on' and 'off' the laser repeatedly during a reaction resulted in significantly higher rates only when the light was 'on', which validated the continuous participation of LSPR excitation in driving the reaction (Fig. 3e). Transient absorption (TA) measurements were performed to verify the excited charge carrier dynamics, where the hot charge carrier generation efficiencies of AuNC, Pd/AuNC, AgNC and Pd/AgNC were sufficiently high to be extracted during the catalytic reactions, and the lifetime constants of all the metal NCs were minimally affected by the pCOL modification of the plasmonic NC surface (Supplementary Figs. 32 and 33). A recyclability test using the same pCOL-Pd/AuNC@$h$SiO$_2$ catalyst multiple times (five cycles) exhibited a negligible loss in activity and selectivity, validating the highly robust nature of the hybrid catalyst, which was also confirmed by the minimal change in catalyst morphology and

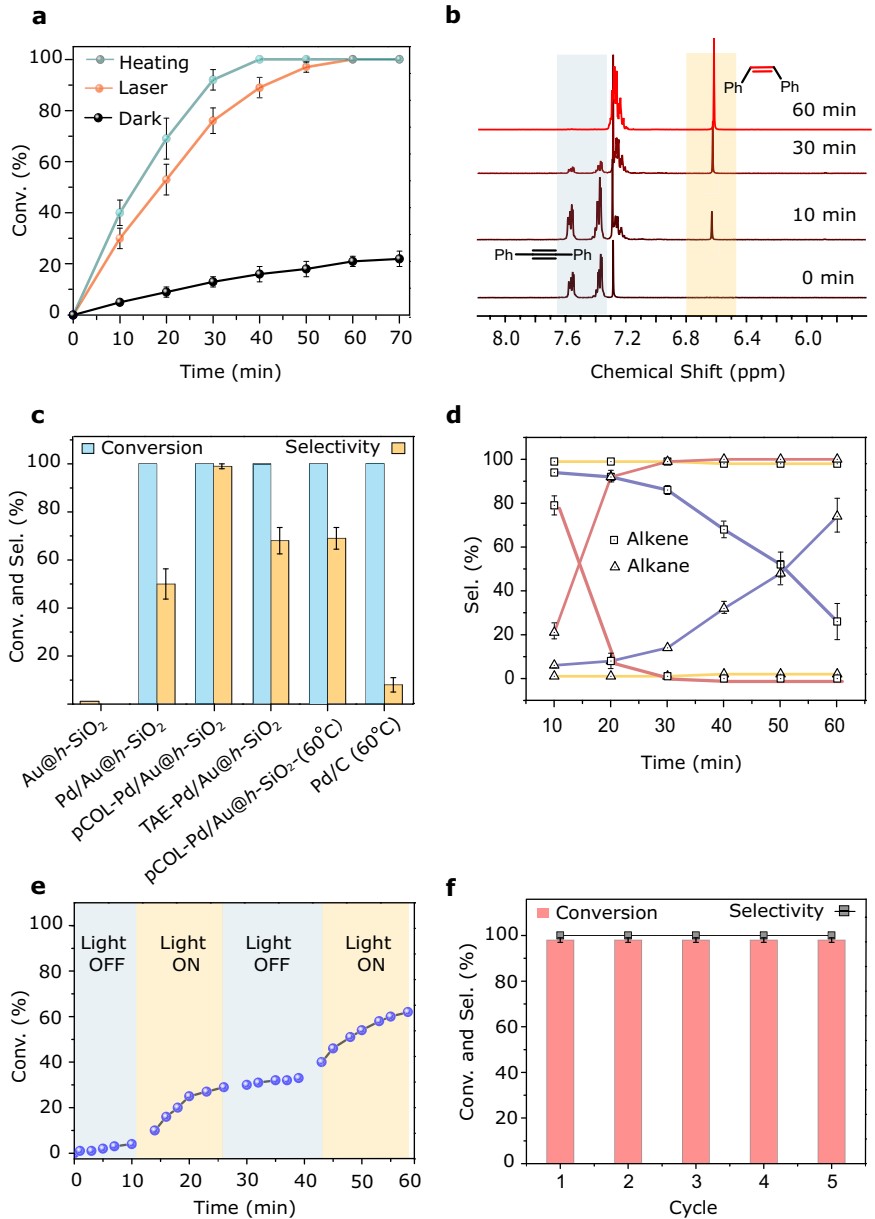

**Fig. 3 | Catalysis data for diphenylacetylene semihydrogenation. a** Reaction kinetics of diphenylethylene (alkene) production using pCOL-Pd/AuNC@*h*-SiO₂ under different conditions. **b** ¹H NMR spectra of the crude reaction mixture at different times using pCOL-Pd/AuNC@*h*-SiO₂ under laser irradiation. **c** Conversion yields and alkene formation selectivities using different catalysts. **d** Time-dependent reaction mixture analysis for the presence of diphenylethylene and diphenylethane in diphenylacetylene hydrogenation catalysed by pCOL-Pd/AuNC@*h*-SiO₂ (yellow), commercial Pd/C (red) and Pd/AuNC@*h*-SiO₂ (blue). **e** Time-dependent conversion yields of semihydrogenation under repetitive light on and off conditions. **f** Recyclability test of pCOL-Pd/AuNC@*h*-SiO₂ for semihydrogenation of diphenylacetylene. Error bars in panels **a**, **c**, **d** and **f** represent standard deviation, *n* = 3 independent replicates.

elemental composition after the reactions. (Fig. 3f and Supplementary Fig. 34). Previously, Camargo et al. demonstrated that LSPR excitation can also increase the selectivity of the semihydrogenation reaction by a charge delocalisation mechanism; however, the selectivity was reduced near reaction completion[38]. Remarkably, in our case, similar nonthermal plasmonic participation in selectivity enhancement can also be expected to some extent, along with the additional favourable contribution by the pCOL maintaining high selectivity until reaction completion. Moreover, the relatively short duration (<1 h) of plasmonically induced reactions reduces the risk of overhydrogenation at ambient bulk temperature. Externally raising the temperature during the laser-induced reaction also reduced the **1b** selectivity up to 65%, further validating the importance of nonthermal conditions (Supplementary Fig. 35).

The ultrathin porous pCOL facilitated the easy entry of the alkyne substrate and the postreaction exit of the alkene product to and from the catalytic sites, maintaining fast diffusion-limited reaction rates. In a control experiment, Pd/AuNC encapsulated inside the bulk-COF shell exhibited extremely slow conversion of disubstituted alkyne **1a** (Supplementary Figs. 15 and 36). This was consistent with the results of previous studies on thick COF/MOF shells. Considering the negligible size differences between alkynes, alkenes and alkanes, a purely molecular size-dependent steric effect at the pCOL-Pd/AuNC interface, which determines the reaction selectivity, is unlikely. Alternatively, owing to the presence of Lewis basic nitrogen atoms and aromatic rings, pCOL could play an electronic role in modifying the molecular adsorption behaviours. Notably, TAE-Pd/AuNC@*h*-SiO₂ with non crosslinked aminated TAE ligands exhibited only a slight improvement

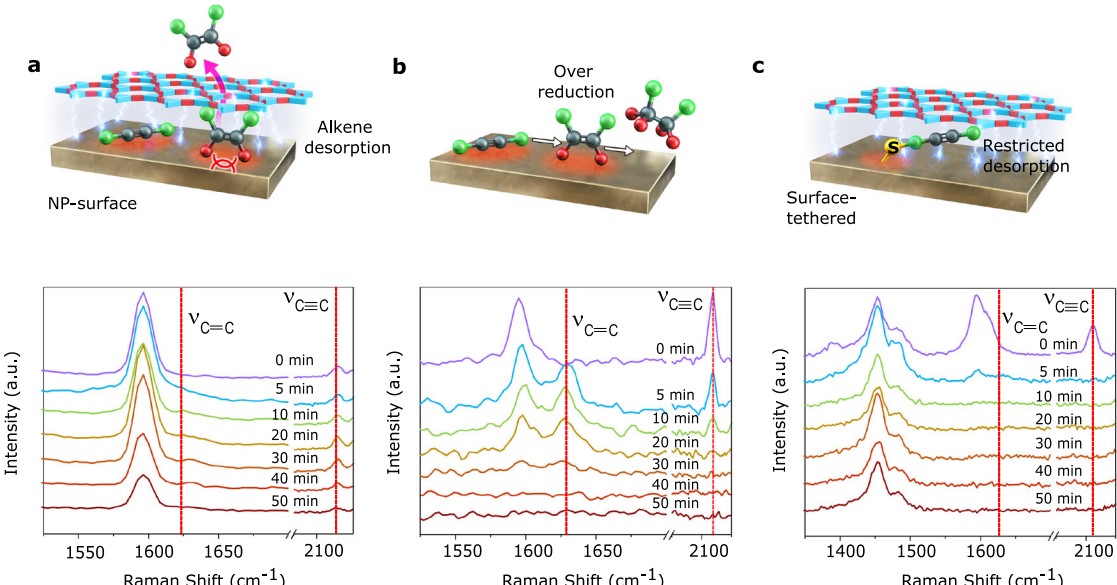

**Fig. 4 | Real-time SERS-based monitoring of diphenylacetylene semihydrogenation using different catalysts. a** Using pCOL-Pd/AuNC@*h*-SiO₂, the Raman peak corresponding to the product alkene emerges only slightly due to the fast desorption from the plasmonic catalytic surface. **b** Using Pd/AuNC@*h*-SiO₂ (without pCOL), the Raman peak corresponding to the product alkene emerges significantly and afterwards vanishes due to the over-reduction. **c** Using pCOL-Pd/AuNC@*h*-SiO₂ (with surface-tethered thiolated diphenylacetylene), the Raman peak corresponding to the product alkene does not appear due to the forced over-reduction of surface-bound alkyne. In molecular models: green = phenyl group, red = H, grey = C.

(<10%) in **1b** selectivity compared to ligand-free Pd/AuNC@*h*-SiO₂ (Supplementary Fig. 37). This signifies the distinctly advantageous role of pCOL over loosely bound molecular ligands. In comparison, the ligand-free commercial Pd/C catalyst at 60 °C exhibited >99% drop in the alkene **1b** selectivity within 1 h (Supplementary Fig. 25). The consistently high TOF at ambient temperature and pressure, along with the minimal loss in product selectivity, is an outcome of the present catalyst design compared to previously reported methods (Supplementary Table 1).

## Mechanistic studies for catalytic reactions

To monitor the real-time molecular adsorption/desorption phenomena during plasmonic catalysis, we conducted an in situ SERS-based study (details in SI and Fig. 4), as SERS signal intensities are highly sensitive to the proximity of probe molecules to the plasmonic surface[46–48]. We performed time-dependent SERS signal acquisition from a solution of alkyne **1a**, NH₃·BH₃ and pCOL-Pd/AuNC (Fig. 4). Originally, the signature Raman peaks at 1595 and 2140 cm⁻¹, corresponding to alkyne **1a**, gradually decreased as the hydrogenation of **1a** to alkene **1b** progressed [the presence of **1b** in the reaction supernatant was confirmed by ¹H-NMR (Supplementary Fig 38)]. This indicated the frequent regular adsorption of alkynes on Pd/AuNC. However, the Raman peak at 1630 cm⁻¹ corresponding to alkene **1b** exhibited only transiently weak presence, indicating the fast desorption from the Pd/AuNC surface in pCOL-Pd/AuNC (Fig. 4a). With the control catalyst devoid of pCOL, alkene **1b** exhibited relatively longer presence (Fig. 4b). In another SERS experiment, we directly attached a thiolated alkyne to the surface of the Pd/AuNC in pCOL-Pd/AuNC (details in SI and Fig. 4c). During the reaction, the SERS signals of the surface-attached alkyne at 1595 and 2140 cm⁻¹ quickly vanished (within 5 min), with the appearance of a peak at 1630 cm⁻¹ corresponding to the double bond for a short time (up to 5 min) owing to the transient stability of the alkene under hydrogenation conditions. In this case, intentionally restricted desorption of the alkyne caused over hydrogenation despite the presence of pCOL. In a separate Raman experiment, we studied the fate of a metal hydride produced from the reaction of NH₃·BH₃ with the catalyst surface, which is a crucial

element in the hydrogenation reaction mechanism. In the case of Pd/AuNC (without pCOL), the Pd-H vibrational frequency at 778 cm⁻¹ was clearly observed from the early stage of the reaction and it decreased slowly with time. In contrast, in the case of pCOL-Pd/AuNC, the Pd-H peak intensity at 778 cm⁻¹ was weak and almost faintly visible from the early stage of the reaction, and thereafter quickly vanished, indicating a more facile hydrogen desorption phenomenon in the case of pCOL-Pd/AuNC compared to Pd/AuNC (Supplementary Fig. 39).

These results directly verified the favourable adsorption of alkynes and fast desorption of the produced alkenes, while maintaining a low effective metal hydride concentration on the surface, as the major events responsible for semihydrogenation selectivity. We propose that the distinct adsorption of alkynes, alkenes and hydrogens on the Pd/AuNC surface is controlled by competitive electronic interactions between perfectly aligned pCOL's crosslinked molecular units. To study the interfacial electronic effect in detail, DFT calculations were performed on a simplified Pd-pCOL interface model using the Vienna ab initio simulation package (VASP) (details in SI and Supplementary Fig. 40). DFT calculations estimated the adsorption energy of the alkene and desorption energy of hydrogen to be −1.45 and 1.56 eV on the Pd-surface modified with pCOL subunit, respectively, which are different than those on the bare Pd surface (Supplementary Figs. 40 and 41). Consistent, extensive coverage of the Pd/AuNC surface by pCOL electronically affects the adsorption energy of C=C double bonds, causing fast desorption. Strong metal–organic electronic interactions between pCOL and Pd/AuNC make the surface electron-rich (confirmed by XPS and Raman spectroscopy) (Supplementary Figs. 42 and 43). DFT-based Bader charge analysis revealed that the Pd surface atoms were more electronically rich (each Pd atom earns 0.38 electrons) after interfacing with the pCOL subunit (Supplementary Fig. 44). These factors favour the adsorption of relatively electrophilic alkynes, but disfavour the adsorption of electron-rich alkenes. Furthermore, we conducted hydrogen/deuterium (H/D)-exchange studies by NMR and concluded that both hydrogens were transferred by BH₃ without any participation from the protic solvent. Additionally, we confirmed a moderate kinetic isotope effect, $k_H/k_D = 1.67$, using NH₃·BD₃ (Supplementary Figs. 45 and 46). Based on this mechanistic

evidence, we illustrate a plausible mechanism in Supplementary Fig. 47. In a substrate-scope study, pCOL-Pd/AuNC demonstrated high conversion yields and consistently high semihydrogenation selectivities for various terminal and internally substituted alkyne substrates (Supplementary Figs. 48–68).

In conclusion, we created and utilised a metal–organic interface to effectively control molecular adsorption–desorption behaviour during heterogeneous catalysis. *Skin-like* ultrathin pCOL formed a microenvironment on the metal surface, endowing favourable steric and electronic effects, allowing facile molecular diffusion and consistently remaining on the NC surface to dictate the reaction selectivity. As a crucial advancement over common organic ligands and thick rigid shells, in a middle-ground approach, our strategy optimally harnesses the exotic features of organic modifiers on the metal NP surface[49–51]. Beyond the rich literature on the effect of metal–metal/metal oxide interfaces, our study opens the avenues for controlling and exploiting metal–organic interfaces for sustainable catalytic synthesis. However, organic modifiers on NC surfaces are commonly borrowed from bottom-up colloidal synthesis routes. The direct synthesis of different metal nanostructures with suitable designer COLs must be explored for wide adoption and diversification.

## Methods
### Synthesis of pCOL/AgNC@*h*-SiO₂
In a 8 mL glass vial, 1,3,5-tris(4-aminophenyl)benzene (TAE) (8.8 mg, 0.025 mmol) and 1,4-benzenedimethanol (DAL) (7 mg, 0.05 mmol) were mixed with a 5 mL solution of 1,4-dioxane:mesitylene (4:1, v/v) using ultrasonication. In another glass vial, AgNC@*h*-SiO₂ (5 mg) was thoroughly dispersed in 1 mL solution of 1,4-dioxane:mesitylene (4:1, v/v) using sonication. This dispersion was quickly injected into the first vial containing the TAE and DAL solution, all while under vigorous stirring with a small (10 × 3 mm) magnetic stirring bar. Next, the entire solution was exposed under a blue laser (405 nm; 0.3 W/cm²) for 1 h with constant stirring at 500 rpm, with details of the instrumental setup as shown in SI, Experimental setup-1. After the reaction was completed, the product was centrifuged and subsequently washed twice with a mixture of 1,4-dioxane:mesitylene (4:1, v/v) followed by anhydrous ethanol (99.9%). The final product, pCOL/AgNC@*h*-SiO₂, was dispersed in anhydrous ethanol (99.9%) and stored at 4 °C.

### Synthesis of pCOL/AuNC@*h*-SiO₂
In the beginning, the mixture of TAE (8.8 mg, 0.025 mmol) and DAL (7 mg, 0.05 mmol) was prepared in 5 ml of 1,4-dioxane:mesitylene (4:1, v/v) solution. A separately prepared solution of AuNC@*h*-SiO₂ (5 mg) in 1 mL of 1,4-dioxane:mesitylene (4:1, v/v) was quickly transferred into the vial containing the TAE and DAL mixture and stirred with a magnetic stirring bar (10 × 3 mm). Next, the solution was irradiated with blue laser (405 nm; 0.3 W/cm²) for an hour with constant stirring at 500 rpm. The product was subjected to centrifugation and underwent a washing process (twice) using a mixture of 1,4-dioxane:mesitylene (4:1, v/v) and anhydrous ethanol (99.9%). The final product, pCOL/AuNC@*h*-SiO₂, was dispersed in anhydrous ethanol (99.9%) and stored at a temperature of 4 °C for further use.

### Synthesis of pCOL-Pd/AuNC@*h*-SiO₂
Initially, 4 mg of pCOL/AuNC@*h*-SiO₂ was dispersed in 5 mL of EG and exposed under a 532 nm laser (0.3 W/cm²), followed by rapid injection of 20 μL Na₂PdCl₄ (5 mM in EG). Then, the entire solution was continuously stirred at 500 rpm for 30 min under laser irradiation using a magnetic stirring bar (10 × 3 mm). Finally, the product was centrifuged and washed twice with 2 mL of anhydrous ethanol (99.9%). The final product, pCOL-Pd/AuNC@*h*-SiO₂, was dispersed in anhydrous ethanol (99.9%) and stored at a temperature of 4 °C for further use.

### Semihydrogenation of diphenylacetylene
To initiate a semihydrogenation reaction, diphenylacetylene (10 mg, 0.056 mmol, 1 equiv.) was dissolved in 1 mL of methanol in a glass vial. Subsequently, the synthesised pCOL-Pd/AuNC@*h*-SiO₂ (1 mg, Pd 0.05 mol%) catalyst was transferred into that solution and fully dispersed under continuous stirring for 5 min. Following this, NH₃.BH₃; (8.6 mg, 0.280 mmol, 5 equiv.) was added to the reaction mixture. The entire solution was irradiated under a 405 nm laser (0.3 W/cm²) with constant stirring at 500 rpm using a small (10 × 3 mm) magnetic stirring bar for the next 1 h. Upon the completion of the reaction, the catalyst was removed by centrifugation (10,000 rpm) and the methanol was evaporated under reduced pressure. Following that, 1 mL of DI-water was added and the product was extracted in 1 mL of ethyl acetate. The ethyl acetate layers were combined, dried over sodium sulphate, and concentrated under reduced pressure, and the ¹H NMR was recorded (conversion 100%; selectivity of alkene ~99%, mass yield of alkene ~9.9 mg). The conversion and selectivity were calculated using the relative peak integration values.

### Real-time SERS study for semihydrogenation
In the beginning, diphenylacetylene (10 mg, 0.056 mmol, 1 equiv.) and pCOL-Pd/AuNC@*h*-SiO₂ catalyst (1 mg, Pd 0.05 mol%) were thoroughly mixed in 1 ml methanol and continuously stirred using a magnetic stirring bar for 5 min. Subsequently, NH₃.BH₃ (8.6 mg, 0.280 mmol, 5 equiv.) was quickly added into the solution and thoroughly mixed. Immediately, a small portion (100 μL) of the reaction mixture was pipetted out and transferred into the micro-holes of the microscopic glass slide, fully covering the hole with an ultra-thin round glass slide (as shown in SI, Experimental setup-3). It was then placed under a 532 nm laser attached to a Raman spectrometer, and the corresponding time-dependent spectra were recorded. A similar approach was employed to obtain time-dependent Raman spectra using the controlled catalyst, Pd/AuNC@*h*-SiO₂. In the case of thiolated-diphenylacetylene tethered on pCOL-Pd/AuNC@*h*-SiO₂ (synthesis details in SI), 1 mg of catalyst was dispersed in 1 mL methanol, followed by a rapid addition of NH₃.BH₃ (8.6 mg, 0.280 mmol, 5 equiv.) under constant stirring. Then, 100 μL of the reaction mixture was pipetted out, and a real-time SERS study was performed.

## Data availability
Source data are provided as a Source Data file. Source data are provided in this paper.

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

## Acknowledgements

This work was supported by the Basic Science Research Programme through the National Research Foundation of Korea (NRF) funded by the Ministry of Science, ICT & Future Planning (MSIP) (Grant NRF-2016R1A3B1907559) (I.S.L.).

## Author contributions

A.K. and I.S.L. conceived the idea, supervised the research and wrote the manuscript with the contribution of all co-authors. A.A. synthesised and characterised the materials and acquired the data. T.B.M. and N.K. performed the organic reactions. Y.Yo., supervised by H.C.C., performed the Raman studies. H.J., supervised by J.W.H., performed DFT calculations. T.J., supervised by Y.P., performed the TA spectroscopy studies. B.S., supervised by Y.Yu., performed the in situ catalysis monitoring.

## Competing interests

The authors declare no competing interests.
