## [Peer Review File · Nature Communications]

REVIEWER COMMENTS

Reviewer #1 (Remarks to the Author):

This MS reports an approach to enhance the selectivity of catalytic reactions by constructing a skin-like few-nm ultrathin crystalline porous covalent organic overlayer (pCOL) directly on the plasmonic NC surface. They prepared this conformal and ultrathin pCOL via plasmonic surface-limited oxidative activation and intermolecular covalent cross-linking of molecular units. This material was then studied for catalytic semi-hydrogenation of substituted alkynes to alkenes.

The catalyst design is just an extension of their similar work in *J. Am. Chem. Soc.* 2021, 143, 28, 10582–10589, instead of metal lamination, now they used pCOL lamination; other than this, there is no novelty in this work. Characterization is thorough, though.

Catalysis studies are not systematic; no stability studies, no comparison with reported literature, no detailed reaction kinetics (only time vs. conversion plots are given). TON and TOF are missing; only conversion can not be the correct way to evaluate the catalysts. These results do not seem better than previously reported.

Catalyst performance analysis by using only NMR seems not correct, E.g. in Figure 3c,3d, at 60 minutes (light), they claimed nearly 100% conversion and 100% selectivity for alkene, but if one sees the NMR very closely (Fig 3b), signals at 7.4 and 7.6 ppm are still there at 60 minutes; it is clearly not 100% and this claim seems incorrect. I think catalysis studies are not conducted properly (GC or GC-MS analysis of the reaction mixture was required).

Although the SERS study (in Figure 4) seems good, it did not provide any information about the reaction mechanism. In-situ DRIFTS/IR is required to understand the molecular mechanism and get insights into how pCOL plays a role (if any). Excited charge dynamics and the role of pCOL are also ignored and not studied at all.

MS is silent about thermal Vs. Non-thermal effect (Fig S19 doesn't really provide any inputs), without which one can not judge the importance and novelty of this work.

The claim of strong metal-organic interactions is not supported by experimental studies.

Overall, I see this as routine work with poor novelty, catalysis studies not conducted properly and lacks deep mechanistic insights. Hence this MS is not suitable for *Nature Communications*.

Reviewer #2 (Remarks to the Author):

This manuscript presents a novel approach to enhance the selectivity of catalytic reactions by

constructing a skin-like few-nm ultrathin crystalline porous covalent organic overlayer (pCOL) directly on the plasmonic NC surface. Conformal and ultrathin pCOL (< 5 nm) is synthesized through plasmonic surface-limited oxidative activation and intermolecular covalent cross-linking of molecular units. Through this method, a light-operated multi-component pCOL interfaced plasmonic-catalytic platform consisting of Pd-modified Au NC inside a hollow silica shell is constructed to achieve highly efficient and selective semi-hydrogenation of substituted alkynes to alkenes. Overall, this work demonstrates a novel design for constructing metal nanocrystal (NC)-organic interfaces and opens the avenues for controlling and exploiting metal-organic interfaces for sustainable catalytic synthesis. So, I would recommend the manuscript for acceptance subjected to revisions that take into account the following concerns.

1) The characteristic C=N peak in the Raman and FTIR spectra has a different wavenumber. More explanations should be given about it.

2) More evidences should be provided to show that the crystalline, porous COF layer is generated as claimed rather than a polymer layer with a disordered structure.

3) A better selectivity can be achieved in a semi-hydrogenation reaction utilizing laser irradiation than under heating conditions. To fully understand the effect of LSPR on this reaction, more experiments are suggested.

4) In the SERS study, the alkyne exhibits a lower Raman shift (1595 cm⁻¹) compared to the alkene (1630 cm⁻¹). The authors might give more explanations about it.

5) More related studies on confining NCs inside the porous framework to enhance the selectivity are suggested to be cited, for examples, *Angew. Chem. Int. Ed.* 2022, 61, e202116396; *Chem* 2021, 7, 686-698; *J. Am. Chem. Soc.* 2022, 144, 17075-17085; etc.

Reviewer #3 (Remarks to the Author):

Review of:

Ultrathin Covalent Organic Overlaying on Metal Nanocrystals for High Selectivity in Plasmon Induced Catalysis

Summary:

This manuscript presents the deposition of an ultrathin crystalline porous covalent organic overlayer (pCOL) on the surface of plasmonic nanocrystals as an avenue to increase the selectivity of alkyne semi-hydrogenation reactions on Pd surfaces. First, they present a synthesis for the deposition of pCOL on plasmonic nanocrystals of different shapes and compositions (Ag, Au). Then, they use H1 NMR to show that pCOL encapsulated Au@Pd nanocubes have higher selectivity for the semi-hydrogenation of various substituted alkynes to alkenes under illumination and compared to control catalysts. Based on in situ SERS and DFT studies, the authors claim that the pCOL organic overlayer modifies the adsorption

strength of alkenes relative to alkynes, facilitating product desorption before semi-hydrogenation occurs. I believe the strategy to improve selectivity by modifying the metal/overlayer interface is novel and exciting. However, I think the manuscript needs some additional data/clarifications before it can be published in Nature Communications. I have outlined my questions/concerns/suggestions below:

- Can the authors comment on the mechanism of the photocatalytic formation of pCOL on Ag vs Au vs Pd surfaces? Using a 405 nm laser excites LSPR in Ag, but excites interband transitions in Au. Do carriers get extracted in either case? Is it heating induced?
- The authors should add the UV vis spectra for AgNC@SiO₂ to Figure S3 for completeness
- On page 6, “In this designed catalytic nanoreactor (pCOL-Pd/AuNC), effect of pCOL on reaction selectivity can be reliably studied: plasmonic AuNC will function as nanoantennae, supplying highly localized energy flow to induce the reaction specifically at the interfacial [Pd]-sites and h-SiO₂ will ensure aggregation-free homogeneous dispersion of the catalyst” should have some citations related to the claims of highly localized energy flow and aggregation prevention.
- Figure 3a, looks like TAE-Pd/AuNC@h-SiO₂ has same conversion and selectivity as pCOL-Pd/AuNC@h-SiO₂ at 60C. Why was 60C chosen for these experiments? How does the conversion and selectivity change at different (lower) temperatures?
- There should be a control experiment included where Pd nanocrystals modified with pCOL and used for the semihydrogenation reaction under light and dark. I believe this control experiment is more relevant than the Pd/C one. It will also yield information related to the importance of the plasmonic nanocrystal core.
- The authors claim the pCOL overlayer changes the binding energy of alkenes relative to alkynes on Pd and that facilitates desorption, which is responsible for the increase in selectivity. Can the authors discuss the viability of other mechanisms where the selectivity might increase (such as charge transfer or photothermal heating)? I bring this up since Figure 1 shows a photoinduced electron traveling from Au to Pd.

Point-by-point Response to the Reviewers' comments

Reviewer #1:

This MS reports an approach to enhance the selectivity of catalytic reactions by constructing a skin-like few-nm ultrathin crystalline porous covalent organic overlayer (pCOL) directly on the plasmonic NC surface. They prepared this conformal and ultrathin pCOL via plasmonic surface-limited oxidative activation and intermolecular covalent cross-linking of molecular units. This material was then studied for catalytic semi-hydrogenation of substituted alkynes to alkenes.

The catalyst design is just an extension of their similar work in J. Am. Chem. Soc. 2021, 143, 28, 10582–10589, instead of metal lamination, now they used pCOL lamination; other than this, there is no novelty in this work. Characterization is thorough, though.

Response: Previously, we had modified ligand-free plasmonic NC surface with a conformal catalytic metal (Pt, Pd, Ru) atomic layer, constructing a “*metal-metal*” hetero-interface for plasmon-induced catalysis. In a clear distinction, current catalyst design consists of a unique “*metal-organic*” interface achieved by controlling the covalent organic molecular cross-linking chemistry. Conventional metal-organic interfaces involve flexible randomly organized ligands or difficult-to-permeate thick shells which are challenging to employ for controlling the reaction selectivity. In a distinct advancement to our previous catalyst-design which was mainly focused on accelerating the reactions, present work addresses the more challenging topic of simultaneously controlling the reaction selectivity during plasmon-induced catalysis. To clarify this important difference, we have added relevant discussion in the revised manuscript (Page 4, line 3-9).

1. Catalysis studies are not systematic; no stability studies, no comparison with reported literature, no detailed reaction kinetics (only time vs. conversion plots are given). TON and TOF are missing; only conversion can not be the correct way to evaluate the catalysts. These results do not seem better than previously reported.

Response: As per the reviewer's suggestion, we have performed catalyst stability test, performance comparison with the relevant literature reports and reaction kinetics by estimating turnover frequency (TOF) values. **pCOL-Pd/AuNC** demonstrated consistently high alkene product yields and selectivities, tested up to 5 cycles with minimal change in the catalyst's morphology and elemental composition, as confirmed by high resolution transmission electron microscopy (HRTEM) and inductively coupled plasma atomic emission spectroscopy (ICP-AES) elemental analysis (Figure 3f, and Supplementary Fig. 34). We have

estimated and compared the TOF values of **pCOL-Pd/AuNC** and **Pd/AuNC** (Supplementary Fig. 26). Although there is not much difference in TOF of **pCOL-Pd/AuNC** (TOF 63 min^{-1}) and **Pd/AuNC** (TOF 60 min^{-1}), selectivity dropped significantly (52 % alkene) in the latter case (catalyst without pCOL). We have also provided reported literature for the comparison under various reaction conditions used for semihydrogenation of alkyne (Supplementary Table 1). We have added the new data, literature comparison table and relevant discussions in the revised manuscript (Page 7, line 25 and Page 9, line 3-6) and Supplementary Information (Supplementary Figs. 26, 34 and Supplementary Table 1).

Supplementary Figure 35. Stability of the catalyst. (a) TEM and (b) HRTEM images of pCOL-Pd/AuNC before and after catalysis. (c) ICP-AES data of Au and Pd weight % before and after the reaction.

Supplementary Figure 26. Turn over frequency (TOF) calculation. TOF (min⁻¹) data and corresponding conversion and selectivity (%) calculated for the semihydrogenation reaction from alkyne (diphenylacetylene) to alkene (stilbene) using pCOL-Pd/AuNC@*h*-SiO₂ and Pd/AuNC@*h*-SiO₂ under the exposure of 405 nm laser (0.3 W/cm²).

SN	Catalyst	TOF (min ⁻¹)	Reaction conditions	Hydrogen Source	Time (min)	Conversion/Selectivity (%)	Ref.
1	pCOL-Pd/AuNC@ h -SiO ₂	63	RT; 1 atm	NH ₃ BH ₃	60	100/99	This work
2	PdCu@Cu ₂ O	0.625	30 °C; 0.1 MPa	H ₂	800	99/95	1
3	Pd-In NPs	6.66	80 °C; 10 bar	H ₂	15	100/85	2
4	Ni-Y Complex	0.025	70 °C; 4.6 atm	H ₂	1440	99/99	3

5	Pd/NHC	0.21	145 °C;	H ₂	120	99/98	4
6	CuPd@ZIF-8	14	RT; 1 atm	NH ₃ BH ₃	5	7/99	5
7	Pd+PEI@HSS	0.55	30 °C; 1atm	H ₂	360	99/95	6
8	Pd ₃ Pb CNCs	0.23	50 °C; 2 atm	H ₂	300	100/92	7
9	Mn Complex	0.083	60-70 °C;	NH ₃ BH ₃	1200	100/99	8
10	Co(II)/NaBH ₄	0.069	RT; 3 bar	H ₂	1440	100/84	9
11	NiCl ₂	49	RT	NaBH ₄	5	98/94	10
12	PdNPore	0.019	RT; 1atm	H ₂	1080	95/97	11
13	Co/phen@SiO ₂ -800	0.11	120 °C; 30 bar	H ₂	900	99/90	12
14	Pd0-AmP-HSN	1.77	RT; 1atm	H ₂	210	93/89	13
15	Pd@Ag-0.20	1.26	RT; 1 atm	H ₂	240	99/99	14
16	AuNPore	0.083	70 °C;	HCOOH	240	100/99	15
17	Pd/IL/MOF	8.6	30 °C; 1atm	H ₂	180	99/99	16
18	Pd-Pb alloy NCs	0.91	RT; 1 atm	H ₂	210	96/99	17
19	Pd/SBA-gt-PEI.	21.7	RT; 0.1 MPa	H ₂	45	98/90	18

Supplementary Table 1. Comparative efficiency in catalysis for semihydrogenation of alkyne. Table contains catalytic efficiencies of reported catalysts compared to pCOL-Pd/AuNC@h-SiO₂ towards the semihydrogenation of alkynes (diphenylacetylene).

2. Catalyst performance analysis by using only NMR seems not correct, E.g. in Figure 3c,3d, at 60 minutes (light), they claimed nearly 100% conversion and 100% selectivity for alkene, but if one sees the NMR very closely (Fig 3b), signals at 7.4 and 7.6 ppm are still there at 60 minutes; it is clearly not 100% and this

claim seems incorrect. I think catalysis studies are not conducted properly (GC or GC-MS analysis of the reaction mixture was required).

Response: As the reviewer has pointed out, we have again conducted the catalysis experiment and collected the ¹H NMR spectra, to confirm the >99% conversion of diphenylacetylene to *cis*-stilbene (Revised Figure 3). Also, we have acquired time-dependent GC-MS data during the course of a reaction (Supplementary Fig. 22). The GC-MS data also confirmed >99% conversion of diphenylacetylene into *cis*-stilbene as a result of semi-hydrogenation. We have added the revised ¹H NMR and new GC-MS data in the manuscript (Page 7, line 15) and Supplementary Information (Supplementary Figs. 21-23).

No	Conc. of Mesitylene (c _i) (M)	Area of Mesitylene (a _i)	Conc. of Diphenylacetylene (c _x) (M)	Area of Diphenylacetylene (a _x)	R _f = (a _x /a _i)/(c _x /c _i)
1	0.03 M	148580387	0.1 M	358165608	0.58664
2	0.03 M	169658485	0.07 M	296190770	
3	0.03 M	169447354	0.05 M	296405612	
4	0.03 M	169530429	0.03 M	179819747	
5	0.03 M	158194984	0.01 M	96209733	

No	Conc. of Mesitylene (c _i) (M)	Area of Mesitylene (a _i)	Conc. of Cis -stilbene (c _x) (M)	Area of Cis -stilbene (a _x)	R _f = (a _x /a _i)/(c _x /c _i)
1	0.03 M	159236050	0.1 M	382695734	0.58846
2	0.03 M	152214857	0.07 M	327690963	
3	0.03 M	161956883	0.05 M	268247803	
4	0.03 M	156543616	0.03 M	211318228	
5	0.03 M	167249591	0.01 M	106068433	

Supplementary Figure 21. Response factor calculation for GC-MS analysis. Determination of response factor (R_f) for diphenylacetylene and *cis*-stilbene.

Supplementary Figure 22. GC analysis for the catalytic semihydrogenation of diphenylacetylene using pCOL-Pd/AuNC@*h*-SiO₂. Time dependent stacked GC chromatographs showing the stepwise evolution of semi-hydrogenated products from alkyne (diphenylacetylene) in presence of mesitylene as internal standard, with consistent high selectivity towards forming alkene (stilbene) as a main product using pCOL-Pd/AuNC@*h*-SiO₂ under the exposure of 405 nm laser (0.3 W/cm²).

Supplementary Figure 23. Mass-spectrometry results. Mass spectrometry data of the chromatographic peaks acquired during the semihydrogenation reaction of alkyne (diphenylacetylene) in presence of mesitylene as internal standard, with consistent high selectivity towards forming alkene (stilbene) as a main product using pCOL-Pd/AuNC@h-SiO₂ under the exposure of 405 nm laser (0.3 W/cm²).

3. Although the SERS study (in Figure 4) seems good, it did not provide any information about the reaction mechanism. In-situ DRIFTS/IR is required to understand the molecular mechanism and get insights into how pCOL plays a role (if any).

Response: We have conducted additional *in situ* monitoring of the reaction using Raman spectroscopy which is more suitable to detect signature molecular vibration frequencies while interacting with the plasmonic catalyst surface. In addition, deuterium labelling NMR experiments gave us further insight into the molecular mechanism in details.

In a time-dependent Raman signal acquisition from a solution of alkyne, NH₃·BH₃ and pCOL-Pd/AuNC (Figure 4): within a short time (< 1 min) alkynes started interacting with the catalyst as indicated by the strong signals at 1595 cm⁻¹ which gradually started decreasing in intensity as the reaction progressed. A new weak intensity peak at 1630 cm⁻¹ corresponding to the product alkene started appearing within 10 min.

Consistent low intensity of the product alkene peak suggested its fast desorption from the catalyst surface. However, in the case of Pd/AuNC catalyst (without pCOL), much higher alkene peak intensity at 1630 cm^{-1} was observed during the course of the reaction, indicating much higher adsorption of alkene on catalyst surface. In a separate *in situ* Raman experiment (Supplementary Fig. S39), we studied the fate of metal hydride produced from the reaction of $\text{NH}_3\cdot\text{BH}_3$ with the catalyst surface, which is a crucial factor in the hydrogenation reaction mechanism. In the case of Pd/AuNC (without pCOL), the Pd-H vibrational frequency at 778 cm^{-1} was clearly observed from early stage of the reaction and it decreased slowly with time. Whereas, in the case of **pCOL-Pd/AuNC**, the Pd-H peak intensity at 778 cm^{-1} was weak and almost faintly visible from the early stage of the reaction and thereafter found to be quickly vanished, indicating more facile hydrogen desorption phenomena in the case of **pCOL-Pd/AuNC** as compared to Pd/AuNC. These comparative *in situ* Raman spectroscopic reaction monitoring revealed crucial role of pCOL in controlling the molecular adsorption behaviors on catalyst surface: pCOL not only causes faster desorption of the alkene product but also keeps the low concentration of reductive hydrides on metal surface. Both of these effects are crucial for avoiding over-hydrogenation of alkene product, leading to the high semi-hydrogenation selectivity. In corroboration, DFT-based adsorption energy calculations showed faster desorption of the alkene and hydrogens on Pd-surface modified with pCOL subunit (Supplementary Figs. 40-41).

Supplementary Figure 39. In-situ Raman data of hydrogen desorption. Time dependent *in-situ* Raman signals corresponding to Pd-H bonding on (a) pCOL-Pd/AuNC (b) Pd/AuNC catalyst's surface.

Supplementary Figure 41. Hydrogen desorption energy calculations and DFT models for interactions of hydrogen molecule on catalytic surface. DFT optimized model structure for hydrogen desorption phenomena from (a) SU-pCOL@Pd and Pd surface respectively. (b) Calculations for the desorption energy of hydrogen from SU-pCOL@Pd and Pd surface.

To get further insight into the molecular mechanism, we conducted hydrogen/deuterium (H/D)-exchange studies by NMR. Replacing CH_3OH by CD_3OD and $\text{NH}_3\cdot\text{BH}_3$ by $\text{ND}_3\cdot\text{BH}_3$ produced no deuterium incorporation. However, replacing $\text{NH}_3\cdot\text{BH}_3$ by $\text{NH}_3\cdot\text{BD}_3$ produced di-deuterated alkene. Also, we confirmed typically moderate isotope effect: $k_{\text{H}}/k_{\text{D}} = 1.67$ using $\text{NH}_3\cdot\text{BD}_3$ (Supplementary Fig. 45 and 46). These results affirmed the direct H-transfer (100%) from BH_3 to alkyne on the metal surface where molecular interaction is controlled by pCOL layer. We have added mechanistic study data and corresponding discussion in the manuscript and SI (Page 10, line 21-26 and Page 11, line 10-12). By taking the evidences from in situ Raman spectroscopy and NMR-based deuterium labelling studies, we have illustrated a plausible mechanism in (Supplementary Fig. 47).

Supplementary Figure 45. Hydrogen/deuterium (H/D)-exchange study and kinetic isotope effects. (a) The time dependent stacked ^1H NMR data for semihydrogenation of diphenylacetylene using NH_3BD_3 in

presence of pCOL-Pd/Au-cube@*h*-SiO₂ catalyst under 405 nm laser. (b) The linear data plot in the box shows typically moderate and comparable kinetic isotope effect: $K_H/K_D = 1.67$ using NH₃BD₃. (c) Results of deuterium exchange reaction study using different deuterated reagent sources.

Supplementary Figure 46. Hydrogen/deuterium (H/D)-exchange study: Crude ¹H NMR data for semihydrogenation of diphenylacetylene with (a) ND₃BH₃ and (b) CD₃OD using pCOL-Pd/Au-cube@*h*-SiO₂ as catalyst under 405 nm laser

Supplementary Figure 47. Plausible mechanism for Semihydrogenation of alkyne. Schematic representations of the semihydrogenation mechanism involved on the (a) pCOL-Pd/AuNC and (b) Pd/AuNC surface.

4. Excited charge dynamics and the role of pCOL are also ignored and not studied at all.

Response: As suggested by the reviewer, we have investigated the generation and dynamics of hot charge-carriers by transient absorption (TA) spectroscopy. Estimation of the $\Delta OD_{\max}/OD$ values from TA spectra can quantify the transient proportions of excited charge-carriers in different plasmonic catalysts; where ΔOD_{\max} is maximum negative change in the optical density at LSPR band and OD is the initial optical density at the steady state, measured against different decay times (Supplementary Figs. 32 and 33). Hot charge-carrier generation efficiency ($\Delta OD_{\max}/OD$) of AuNC and Pd/AuNC were found to be minimally affected by the pCOL modification. We also estimated the hot charge-carrier's life time constant (τ) by measuring the transient $\Delta OD/OD$ kinetics at picoseconds timescale. The τ values for AuNC, Pd/AuNC, pCOL-AuNC and pCOL-Pd/AuNC were estimated to be 1.13 ± 0.02 , 0.47 ± 0.01 , 1.1 ± 0.02 and 0.45 ± 0.01 ps respectively (Supplementary Figs. 32). Also, similar trends of excited charge dynamics was observed in the case of AgNC, Pd/AgNC, pCOL-AgNC and pCOL-Pd/AgNC such as, 4.2 ± 0.02 , 2.51 ± 0.05 , 3.8 ± 0.3 and 2.1 ± 0.01 ps respectively. Evidently, hot charge-carrier generation efficiency and life-time constant of the plasmonic catalysts were minimally influenced by pCOL modification, which is a crucial requirement

in plasmonic catalysis. We have added the TA spectroscopy data and relevant discussion in the revised manuscript (page 8, line 26-27 and Page 1-2) and SI (Supplementary Fig. 32-33).

Supplementary Figure 32. Transient absorption (TA) analysis. Excited charge carrier dynamics data showing the life time, TA-data plot corresponds to charge carrier generation efficiencies and TA-contour plot for (a,b and c) AuNC, (d,e and f) Pd/AuNC, (g,h and i) pCOL-AuNC, (j,k and l) pCOL-Pd/AuNC respectively.

Supplementary Figure 33. Transient absorption (TA) analysis. Excited charge carrier dynamics data showing the life time, TA-data plot corresponds to charge carrier generation efficiencies and TA-contour plot for (a,b and c) AgNC, (d,e and f) Pd/AgNC, (g,h and i) pCOL-AgNC, (j,k and l) pCOL-Pd/AgNC respectively.

5. MS is silent about thermal Vs. Non-thermal effect (Fig S19 doesn't really provide any inputs), without which one can not judge the importance and novelty of this work.

Response: Under laser irradiation, pCOL-Pd/AuNC resulted >99% conversion within 1h at bulk solution temperature < 30 °C. In contrast, maintaining identical reaction temperature under dark-conditions could only result < 20 % conversion; and raising the bulk solution temperature > 60 °C could achieve >99% conversion. Such stark influence of laser irradiation suggested LSPR-induced catalytic mechanism where plasmonic energy can be efficiently transferred to surface catalytic sites.¹⁻³ To further understand the difference between thermal and laser irradiation conditions, we studied the effect of externally raising bulk reaction temperature on alkene product selectivity under dark-conditions (Supplementary Fig. 27). At lower temperatures (< 40 °C) when conversions were low (< 20%), high alkene selectivity (up to >99%) was monitored. However, further raising the temperatures to get higher conversions, caused dramatic loss in alkene selectivity (up to < 5% at 80 °C). In a separate study under laser irradiation, increasing the laser flux had much lower adverse effect on alkene product selectivity while high conversions were observed even at low laser fluxes (Supplementary Fig. 29). While externally raising the bulk solution temperature under light irradiation again caused significant loss in alkene product selectivity (Supplementary Fig. S30). Distinctly, under laser irradiation conditions – high conversions in conjunction with high alkene product selectivity – strongly suggested the favorable contribution of non-thermal effects to have major contribution. We have included the new data with mechanistic details in the revised manuscript (page 8, line 2-11) and SI (Supplementary Figs. 27-30).

To further detangle the role of excited charge-carriers in plasmonic catalysis, we performed additional mechanistic experiments. First, we verified the generation and sufficient life-times of hot electrons from TA spectroscopy,^{4,5} as described in the response to previous comment (Supplementary Figs. 32 and 33). As previously proposed, in a colloidal medium under continuous-wave laser excitation at the low intensity, the temperature at the surface of the photoexcited plasmonic catalyst is expected to be nearly the same as that of the medium.⁶ To further estimate the exact temperature on plasmonic catalyst's surface, we carried out a Raman thermometry analysis based on the shift in Stokes and anti-Stokes signals of Raman probe directly modified on catalyst surface (details in Supplementary Information, Supplementary Fig. 31).⁷ This direct experimental estimation suggested that the low continuous wave laser fluxes were not able to increase the surface temperature more than 50 °C. This data again indicated that plasmonic non-thermal effects played dominant role in catalysis induction, rather than contribution from photothermal pathway. Using pure thermal conditions, much higher solution temperatures (>60 °C) were required to achieve reaction rates as high as in the case of laser irradiation which also caused the loss in alkene product selectivity. The results

in our case indicated the dominant role of plasmonic hot charge-carriers (non-thermal pathway) in the reaction rate enhancement. However, depending on the light fluxes and catalyst designs, mixed roles of hot charge-carriers and photothermal effects in different catalytic reactions have also been previously proposed.⁸⁻¹¹ We have included the new data with mechanistic details in the revised manuscript (page 8, line 12-18, 26-27 and Page 9, line 1-3) and Supplementary Information (Supplementary Figs. 31-33).

Supplementary Figure 27. Catalysis under external heating condition. (a) Alkene selectivity plot and (b) the corresponding ¹HNMR data of variable selectivities of the alkene product from alkyne (diphenylacetylene) using pCOL-Pd/AuNC@h-SiO₂ by raising the reaction temperatures from 30 to 80 °C.

Supplementary Figure 29. Catalysis under variable laser flux. (a) Selectivity plot of the alkene product from alkyne (diphenylacetylene) using pCOL-Pd/AuNC@*h*-SiO₂ sequentially increasing the 405 nm laser flux. (b) Corresponding ¹HNMR data with sequential increase in the laser flux.

Supplementary Figure 30. Catalysis external heating during the laser irradiation. (a) Alkene selectivity plot and (b) the corresponding ¹H NMR data of variable selectivities of the alkene product from alkyne (diphenylacetylene) using pCOL-Pd/AuNC@*h*-SiO₂ with sequential increase in external heating external heat up to 60 °C during the laser irradiation (405 nm; 0.3 W/cm²).

Supplementary Figure 31. Raman thermometry analysis. (a) SERS signals (stokes and anti-stokes) of 4-aminothiophenol tethered on Pd/AuNCs surface. (b) Temperatures on plasmonic catalyst's (Pd/AuNC) surface under the irradiation of continuous wave resonant laser.

6. The claim of strong metal-organic interactions is not supported by experimental studies.

Response: We found the evidence of strong metal-organic electronic interaction in **pCOL-Pd/AuNC** by XPS and Raman spectroscopy experiments. In a comparative XPS analysis, deconvoluted Pd 3d_{5/2} peak (at 335.18 eV) in Pd/AuNC shifted to lower binding energy by 0.58 eV after pCOL modification in the case of **pCOL-Pd/AuNC** due to the effective charge transfer from electron-rich pCOL to the d-orbitals of Pd (Supplementary Fig. 42). Metal-organic interaction was further supported by Raman spectroscopy measurements: characteristic Raman peak at 1570 cm⁻¹ corresponding to C=N in the case of **pCOL-Pd/AuNC** was blue shifted as compared to the bulk-COF (1594 cm⁻¹) (Supplementary Fig. 43). We have added the new data and relevant discussion in the revised manuscript (Page 11, line 14-15) and Supplementary Information (Supplementary Figs. 42 and 43).

Supplementary Figure 43. Comparative Raman spectra of pCOL layer with bulk COF. Comparative Raman spectra of pCOL layer showing imine (C=N) peak shifted from bulk COF.

Overall, I see this as routine work with poor novelty, catalysis studies not conducted properly and lacks deep mechanistic insights. Hence this MS is not suitable for Nature Communications.

Response: Although, number of studies have emerged on different plasmonic catalysts for inducing the rate of reactions, simultaneously, controlling the product selectivity is still challenging. Conventional flexible ligands and rigid thick metal-organic/inorganic shells often uncontrollably occupy the catalytic surface and complicate the catalytic mechanisms. In the present work, we solve this bottleneck by

developing a strategy to controllably modify metal NC with a *skin-like* pCOL. Conformal ultrathin pCOL porous jacket around NC renders optimum microenvironment over catalytic sites to reliably control the molecular interaction behavior and in turn reaction selectivity, without compromising the reaction rates. In a proof-of-concept application: our strategy overcomes the commonly observed over-reduction side-reaction during semi-hydrogenation of alkynes. We have conducted additional sets of experiments and accordingly revised the manuscript. Beyond the rich literature on the effect of metal-metal/metal oxide hetero-interfaces, our work opens the avenues for effectively controlling and exploiting metal-organic interfaces for selective catalytic synthesis.

Reviewer #2:

This manuscript presents a novel approach to enhance the selectivity of catalytic reactions by constructing a skin-like few-nm ultrathin crystalline porous covalent organic overlayer (pCOL) directly on the plasmonic NC surface. Conformal and ultrathin pCOL (< 5 nm) is synthesized through plasmonic surface-limited oxidative activation and intermolecular covalent cross-linking of molecular units. Through this method, a light-operated multi-component pCOL interfaced plasmonic-catalytic platform consisting of Pd-modified Au NC inside a hollow silica shell is constructed to achieve highly efficient and selective semi-hydrogenation of substituted alkynes to alkenes. Overall, this work demonstrates a novel design for constructing metal nanocrystal (NC)-organic interfaces and opens the avenues for controlling and exploiting metal-organic interfaces for sustainable catalytic synthesis. So, I would recommend the manuscript for acceptance subjected to revisions that take into account the following concerns.

1. The characteristic C=N peak in the Raman and FTIR spectra has a different wavenumber. More explanations should be given about it.

Response: The characteristic Raman vibrational frequencies are highly sensitive to various parameters including charge-transfer on metal surface which has been extensively studied in surface-enhanced Raman scattering (SERS) effect.^{12,13} In the present case, the strong metal-organic interaction between pCOL and Pd/AuNC surface (also verified by XPS and Raman spectroscopy, Supplementary Fig. 43) may be responsible for the difference in the C=N peak wavenumbers. Metal-organic interaction was further supported by the fact that characteristic Raman peak at 1570 cm⁻¹ corresponding to C=N in the case of

pCOL-Pd/AuNC was blue shifted as compared to the bulk-COF (1594 cm^{-1}) (Supplementary Fig. 43). We have added the plausible explanation in the revised manuscript (Page 11, line 14-15).

Supplementary Figure 43. Comparative Raman spectra of pCOL layer with bulk COF. Comparative Raman spectra of pCOL layer showing imine (C=N) peak shifted from bulk COF.

2. *More evidences should be provided to show that the crystalline, porous COF layer is generated as claimed rather than a polymer layer with a disordered structure.*

Response: As suggested by the reviewer, we attempted to acquire XRD data to unambiguously characterize crystalline nature of porous pCOL layer on metal surface. Despite the crystalline nature of pCOL clearly characterized by HRTEM images (Figure 2), XRD peaks for COF were below detection limit in the powdered sample. This may be due to the multicomponent **pCOL-Pd/AuNC** structure having dominant presence of outer amorphous silica and different interfacing metals accommodating only very thin (*ca.* 2 nm) low content COF layer. To overcome this challenge, we synthesized thicker pCOL (*ca.* 20 nm) on an exposed Pd/AuNP@SiO₂ (designated as pCOL-Pd/AuNP@SiO₂) surface following our laser-induced protocol (details in Supplementary Information, Supplementary Fig. 17). XRD data of pCOL-Pd/AuNP@SiO₂ showed characteristic peaks corresponding to the COF layer at lower angles 2.79 (100), 4.74 (110) and 7.50 (210) (Supplementary Fig. 18). The Brunauer-Emmett-Teller surface area analysis (BET) data of pCOL-Pd/AuNP@SiO₂ confirmed the typical porous nature (pore size 4 nm) of the pCOL layer (Supplementary Fig. 19). In addition, pCOL-Pd/AuNP@SiO₂ demonstrated catalytic

semihydrogenation of diphenylacetylene under laser irradiation resulting to the alkene as major product with 99% selectivity, demonstrating similar effect of COF overlayer in controlling the reaction selectivity.

Supplementary Figure 17. Step-wise synthesis and characterization of pCOL-Pd/AuNP@SiO₂. TEM image of (a) AuNP@SiO₂ (b) Pd/AuNP@SiO₂ and (c) pCOL-Pd/AuNP@SiO₂ and their corresponding EDS-based weight and atomic% of N, C, Au and Pd in Table. ¹H NMR data shows the catalytic semihydrogenation of diphenylacetylene using pCOL-Pd/AuNP@SiO₂ under laser irradiation resulting to the alkene as major product with 99% selectivity.

Supplementary Figure 18. XRD data of pCOL-Pd/AuNP@SiO₂. The XRD data of pCOL-Pd/AuNP@SiO₂ before and after the deposition of pCOL layer to verify the crystallinity of pCOL layer

Supplementary Figure 19. Porous structure analysis of pCOL-Pd/AuNP@SiO₂ (a) N₂ adsorption-desorption isotherm profile of Pd/AuNP@SiO₂ (BET surface area: 10 m²/g) and pCOL-Pd/AuNP@SiO₂ (BET surface area: 19 m²/g) before and after pCOL deposition respectively. (b) Pore size distribution (average pore size 4 nm) of pCOL layer in pCOL-Pd/AuNP@SiO₂.

3. A better selectivity can be achieved in a semi-hydrogenation reaction utilizing laser irradiation than under heating conditions. To fully understand the effect of LSPR on this reaction, more experiments are suggested.

Response: Under laser irradiation, **pCOL-Pd/AuNC** resulted >99% conversion within 1h at bulk solution temperature < 30 °C. In contrast, maintaining identical reaction temperature under dark-conditions could only result < 20 % conversion; and only raising the bulk solution temperature > 60 °C could achieve >99% conversion. Such stark influence of laser irradiation suggested LSPR-induced catalytic mechanism where plasmonic energy can be efficiently transferred to surface catalytic sites. To further understand the difference between thermal and laser irradiation conditions, we studied the effect of externally raising bulk reaction temperature on alkene product selectivity under dark-conditions (Supplementary Figure 27). At lower temperatures (< 40 °C) when conversions were low (< 20%), high alkene selectivity (up to >99%) was monitored. However, further raising the temperatures to get higher conversions, caused dramatic loss in alkene selectivity (up to < 5% at 80 °C). In a separate study under laser irradiation, increasing the laser flux had much lower adverse effect on alkene product selectivity while high conversions were observed even at low laser fluxes (Supplementary Figure 29). While externally raising the bulk solution temperature under light irradiation again caused significant loss in alkene product selectivity (Supplementary Figure 30). Distinctly, under laser irradiation conditions – high conversions in conjunction with high alkene product selectivity – strongly suggested the favorable contribution of non-thermal effects to have major contribution. We have included the new data with mechanistic details in the revised manuscript (page 8, line 2-11) and SI (Supplementary Figs. 27-30).

To further detangle the role of excited charge-carriers in plasmonic catalysis, we performed additional mechanistic experiments. First, we verified the generation and sufficient life-times of hot electrons from TA spectroscopy (Supplementary Figs. 32 and 33).^{5,14} As previously proposed, in a colloidal medium under continuous-wave laser excitation at the low intensity, the temperature at the surface of the photoexcited plasmonic catalyst is expected to be nearly the same as that of the medium.⁶ In our case, to further estimate the exact temperature on plasmonic catalyst's surface, we carried out a Raman thermometry analysis based on the shift in Stokes and anti-Stokes signals of Raman probe directly modified on catalyst surface (details in Supplementary Information, Supplementary Fig. 31).⁷ This direct experimental estimation suggested that the low continuous wave laser fluxes were not able to increase the surface temperature more than 50 °C. This data again indicated that plasmonic non-thermal effects played dominant role in catalysis induction, rather than contribution from photothermal pathway. Using pure thermal conditions, much higher solution

temperatures ($>60\text{ }^{\circ}\text{C}$) were required to achieve reaction rates as high as in the case of laser irradiation which also caused the loss in alkene product selectivity. The results in our case indicated the dominant role of plasmonic hot charge-carriers (non-thermal pathway) in the reaction rate enhancement. However, depending on the light fluxes and catalyst designs, mixed roles of hot charge-carriers and photothermal effects in different catalytic reactions have also been previously proposed.⁸⁻¹¹ We have included the new data with mechanistic details in the revised manuscript (page 8, line 12-18, 26-27 and page 9, line 1-3) and SI (Supplementary Figs. 31-33).

Supplementary Figure 27. Catalysis under external heating condition. (a) Alkene selectivity plot and (b) the corresponding ^1H NMR data of variable selectivities of the alkene product from alkyne (diphenylacetylene) using pCOL-Pd/AuNC@*h*-SiO₂ by raising the reaction temperatures from 30 to 80 $^{\circ}\text{C}$.

Supplementary Figure 29. Catalysis under variable laser flux. (a) Selectivity plot of the alkene product from alkyne (diphenylacetylene) using pCOL-Pd/AuNC@*h*-SiO₂ sequentially increasing the 405 nm laser flux. (b) Corresponding ¹H NMR data with sequential increase in the laser flux.

Supplementary Figure 30. Catalysis external heating during the laser irradiation. (a) Alkene selectivity plot and (b) the corresponding ¹H NMR data of variable selectivities of the alkene product from alkyne (diphenylacetylene) using pCOL-Pd/AuNC@*h*-SiO₂ with sequential increase in external heating external heat up to 60 °C during the laser irradiation (405 nm; 0.3 W/cm²).

Supplementary Figure 31. Raman thermometry analysis. (a) SERS signals (stokes and anti-stokes) of 4-aminothiophenol tethered on Pd/AuNCs surface. (b) Temperatures on plasmonic catalyst's (Pd/AuNC) surface under the irradiation of continuous wave resonant laser.

Supplementary Figure 32. Transient absorption (TA) analysis. Excited charge carrier dynamics data showing the life time, TA-data plot corresponds to charge carrier generation efficiencies and TA-contour plot for (a,b and c) AuNC, (d,e and f) Pd/AuNC, (g,h and i) pCOL-AuNC, (j,k and l) pCOL-Pd/AuNC respectively

Supplementary Figure 33. Transient absorption (TA) analysis. Excited charge carrier dynamics data showing the life time, TA-data plot corresponds to charge carrier generation efficiencies and TA-contour plot for (a,b and c) AgNC, (d,e and f) Pd/AgNC, (g,h and i) pCOL-AgNC, (j,k and l) pCOL-Pd/AgNC respectively.

4. In the SERS study, the alkyne exhibits a lower Raman shift (1595 cm^{-1}) compared to the alkene (1630 cm^{-1}). The authors might give more explanations about it.

Response: For diphenylacetylene, we have monitored two peaks at 2140 cm^{-1} (for $\text{C}\equiv\text{C}$ stretch.) and 1595 cm^{-1} (phenyl vibration mode). Both of these peaks underwent gradual decrease in the intensity as the reaction progressed. For cis-stilbene, we have monitored the peak at 1630 cm^{-1} for $\text{C}=\text{C}$ stretch. We have now added extend SERS spectra in the revised Figure 4 to avoid any confusion.

Fig. 4. Real-time SERS-based monitoring of diphenylacetylene semi-hydrogenation using different catalysts. **a**, using pCOL-Pd/AuNC@ $h\text{SiO}_2$ Raman peak corresponding to the product alkene emerges only slightly due to the fast desorption from the plasmonic catalytic surface. **b**, using Pd/AuNC@ $h\text{SiO}_2$ (without pCOL) Raman peak corresponding to the product alkene emerges significantly and afterwards vanishes due to the over-reduction. **c**, using pCOL-Pd/AuNC@ $h\text{SiO}_2$ (with surface tethered thiolated diphenylacetylene) Raman peak corresponding to the product alkene doesn't appear due to the forced over-reduction of surface-bound alkyne. In molecular models: Green = phenyl group, red = H, Grey = C.

5. More related studies on confining NCs inside the porous framework to enhance the selectivity are

suggested to be cited, for examples, Angew. Chem. Int. Ed. 2022, 61, e202116396; Chem 2021, 7, 686-698; J. Am. Chem. Soc. 2022, 144, 17075-17085; etc.

Response: We have cited the suggested references in the revised Introduction part of the manuscript (Ref# 27, 33, 34).

Reviewer #3:

This manuscript presents the deposition of an ultrathin crystalline porous covalent organic overlayer (pCOL) on the surface of plasmonic nanocrystals as an avenue to increase the selectivity of alkyne semi-hydrogenation reactions on Pd surfaces. First, they present a synthesis for the deposition of pCOL on plasmonic nanocrystals of different shapes and compositions (Ag, Au). Then, they use ¹H NMR to show that pCOL encapsulated Au@Pd nanocubes have higher selectivity for the semi-hydrogenation of various substituted alkynes to alkenes under illumination and compared to control catalysts. Based on in situ SERS and DFT studies, the authors claim that the pCOL organic overlayer modifies the adsorption strength of alkenes relative to alkynes, facilitating product desorption before semi-hydrogenation occurs. I believe the strategy to improve selectivity by modifying the metal/overlayer interface is novel and exciting. However, I think the manuscript needs some additional data/clarifications before it can be published in Nature Communications. I have outlined my questions/concerns/suggestions below:

1. Can the authors comment on the mechanism of the photocatalytic formation of pCOL on Ag vs Au vs Pd surfaces? Using a 405 nm laser excites LSPR in Ag, but excites interband transitions in Au. Do carriers get extracted in either case? Is it heating induced?

Response: For mechanistic study, we have investigated the generation and dynamics of hot charge-carriers by transient absorption (TA) spectroscopy (details in Supplementary Information, Figs. 32 and 33). We estimated the hot charge-carrier's life time constant (τ) for AuNC, Pd/AuNC to be 1.13 ± 0.02 , 0.47 ± 0.01 ps respectively. Also, similar trends of excited charge dynamics were observed in the case of AgNC and Pd/AgNC. Photo-generated hot charge-carriers with sufficient life-times on the plasmonic NC surface could be easily extracted for photocatalytic oxidation of 1,4-benzenedimethanol (DAL) to 1,4-benzenedicarboxyaldehyde (DAE) which can further condense with 1,3,5-tris(4-aminophenyl)benzene (TAE) to form covalently cross-linked pCOL (Figure 2b and Supplementary Fig. 15a). Previously, light induced photochemical oxidation of primary alcohols to aldehydes has been reported by the involvement

of excited charge-carriers. We estimated the exact temperature on plasmonic catalyst's surface, by a Raman thermometry analysis based on the shift in Stokes and anti-Stokes signals of Raman probe directly modified on metal NP surface (details in Supplementary Information, Supplementary Fig. 31). This direct experimental estimation suggested that the low continuous wave laser fluxes were not able to increase the surface temperature more than 50 °C. Accordingly, we attempted pCOL formation on AgNC under heating (at 60 °C) which was unsuccessful (Supplementary Fig. 31 and We have included the new data with mechanistic details in the revised manuscript in page 8, line 12-18, 26-27 and page 9, line 1-3) Using the near infrared (NIR) laser wavelength (808 nm) or deoxygenated solutions containing DAL and TAE did not result any reaction with AgNC@*h*-SiO₂. In a separate experiment, we confirmed the photochemical oxidative role of Ag-surface in conducting polymerization of methyl methacrylate (MMA) resulting to a shell of poly-MMA around AgNC, confirming the extraction of charge-carriers for photochemical reactions. Based on these results, we propose pCOL formation to follow photochemical pathway rather than thermal one. In our experiments, we have used 405 nm laser to induce the formation of pCOL, which is applicable for both AgNC (LSPR resonant) and AuNC (interband transition); however, 532 nm laser (LSPR resonant) could also be applied in the case of AuNC giving similar results (Supplementary Fig. 14). We have added the detailed mechanistic discussion on pCOL formation in the revised manuscript (page 6, line 14-18).

Supplementary Figure 32. Transient absorption (TA) analysis. Excited charge carrier dynamics data showing the life time, TA-data plot corresponds to charge carrier generation efficiencies and TA-contour plot for (a,b and c) AuNC, (d,e and f) Pd/AuNC, (g,h and i) pCOL-AuNC, (j,k and l) pCOL-Pd/AuNC respectively.

Supplementary Figure 33. Transient absorption (TA) analysis. Excited charge carrier dynamics data showing the life time, TA-data plot corresponds to charge carrier generation efficiencies and TA-contour plot for (a,b and c) AgNC, (d,e and f) Pd/AgNC, (g,h and i) pCOL-AgNC, (j,k and l) pCOL-Pd/AgNC respectively.

Supplementary Figure 31. Raman thermometry analysis. (a) SERS signals (stokes and anti-stokes) of 4-aminothiophenol tethered on Pd/AuNCs surface. (b) Temperatures on plasmonic catalyst's (Pd/AuNC) surface under the irradiation of continuous wave resonant laser.

Supplementary Figure 14. pCOL deposition on AuNC@*h*-SiO₂ under irradiation of 532 nm laser.

TEM image of pCOL-AuNC@*h*-SiO₂ having conformal p-COL deposition on AuNCs under the irradiation of 532 nm laser.

2. The authors should add the UV vis spectra for AgNC@SiO₂ to Figure S3 for completeness.

Response: We have added the UV-vis spectra AgNC@SiO₂ in the revised manuscript (Supplementary Fig. 4).

Supplementary Figure 4. Characterizations of AgNC@*h*-SiO₂. UV-vis spectra of AgNCs (black), AgNC@SiO₂ (green), and AgNC@*h*-SiO₂ (Red).

3. On page 6, “In this designed catalytic nanoreactor (pCOL-Pd/AuNC), effect of pCOL on reaction selectivity can be reliably studied: plasmonic AuNC will function as nanoantennae, supplying highly localized energy flow to induce the reaction specifically at the interfacial [Pd]-sites and h-SiO₂ will ensure aggregation-free homogeneous dispersion of the catalyst” should have some citations related to the claims of highly localized energy flow and aggregation prevention.

Response: As suggested by the reviewer, we have cited the relevant references (ref# 35-39) .

4. Figure 3a, looks like TAE-Pd/AuNC@h-SiO₂ has same conversion and selectivity as pCOL-Pd/AuNC@h-SiO₂ at 60C. Why was 60C chosen for these experiments? How does the conversion and selectivity change at different (lower) temperatures?

Response: At 60 °C under dark-conditions, both catalysts afforded similar reasonably high reaction conversion rates but resulted loss in product selectivity (Figure 3c). As the product selectivity changes with the reaction progress (Figure 3d), it will be fair to compare the performances at the enough high temperature (60 °C) giving similar high conversions (> 99%), ideally upon completion of the reaction. At lower temperatures, reaction was much slower giving < 50% conversions even after prolonging the reaction for long times, which is not suitable for selectivity comparisons; and only raising the bulk solution temperature > 60 °C could achieve >99% conversion. We studied the effect of externally raising bulk reaction temperature on alkene product selectivity under dark-conditions (Supplementary Fig. 27). At lower temperatures when conversions were low (< 20%), high alkene selectivity (up to >99%) was monitored. However, further raising the temperatures to get higher conversions, caused dramatic loss in alkene selectivity. We have added this data and relevant discussion in the revised manuscript (page 8, line 2-11).

Supplementary Figure 27. Catalysis under external heating condition. (a) Alkene selectivity plot and (b) the corresponding ^1H NMR data of variable selectivities of the alkene product from alkyne (diphenylacetylene) using pCOL-Pd/AuNC@*h*-SiO₂ by raising the reaction temperatures from 30 to 80 °C.

5. There should be a control experiment included where Pd nanocrystals modified with pCOL and used for the semihydrogenation reaction under light and dark. I believe this control experiment is more relevant than the Pd/C one. It will also yield information related to the importance of the plasmonic nanocrystal core.

Response: As suggested by the reviewer, we have synthesized the control catalyst having Pd NCs modified with pCOL, designated by pCOL-Pd@SiO₂ (details in Supplementary Information, Supplementary Fig. 24). Under laser irradiation, pCOL-Pd@SiO₂ afforded much lower conversions (< 16%) in the absence of any plasmonic assistance from the Au-component. Under dark conditions, performances of pCOL-Pd@SiO₂ and pCOL-Pd/AuNC were similar (Supplementary Fig. 25). Importantly, these control experiments suggested crucial role of plasmonic component (Au) in the catalytic performance. We have added new data in the revised manuscript (Page 7, line 17-20).

Supplementary Figure 24. Characterization of thick pCOL-PdNP@SiO₂. TEM image showing (a) PdNP@SiO₂ and (b) after the deposition of thick pCOL layer as PdNP@SiO₂

Supplementary Figure 25. Controlled catalytic reaction with commercial Pd/C and thick pCOL--PdNP@SiO₂. Time-dependent stacked ¹H NMR data showing the over-hydrogenation of diphenylacetylene using the (a) commercial Pd/C under 60 °C temperature and (b) thick pCOL-PdNP@SiO₂ under light and dark (60 °C, 90 min) condition.

6. The authors claim the pCOL overlayer changes the binding energy of alkenes relative to alkynes on Pd and that facilitates desorption, which is responsible for the increase in selectivity. Can the authors discuss the viability of other mechanisms where the selectivity might increase (such as charge transfer or photothermal heating)? I bring this up since Figure 1 shows a photoinduced electron traveling from Au to Pd.

Response: As suggested by the reviewer, we have added the discussion on the viability of different mechanisms where the selectivity might increase from plasmonic participation (Page 1, line 3-6).

We propose a combined participation of plasmonic effect and pCOL to induce the selectivity of hydrogenation reaction: (i) Role of pCOL: coverage of Pd/AuNC surface by pCOL electronically affects the adsorption energy of C=C double bonds, causing the faster desorption of alkene. (ii) Role of plasmonic effect: LSPR induction can participate in three different ways to affect the reaction pathways (a) electromagnetic field (non-thermal), (b) charge-carriers (non-thermal) (c) photo-thermal heating.

All these factors (plasmonic effect) can be favorable to enhance the reaction rates. But, control of selectivity of semi-hydrogenation depends on two possible pathways guided by *non-thermal* mechanisms: (i) increased reactivity of alkyne as compared to the alkene by favorable participation of charge-carriers depending on the preferable molecular orientation and density of states (DOS) on the metal surface;¹⁴ (ii) NP surface electromagnetic field can have preferable interaction with more polar substrate or intermediate,^{15,16} (iii) faster desorption of H₂, thus limiting the availability of effective concentration of on-surface metal hydrides for additional hydrogenation of alkene.^{16,17} However, in the absence of pCOL, in situ SERS study revealed the higher retention and reduction of alkene over the reaction time-course indicating that the plasmonic effect solely is not enough capable to alter the adsorption energies. Increasing the temperature of the reaction (dark conditions) showed higher rates but adverse effect on selectivity, meaning photothermal heating (if any) cannot increase the selectivity of the reaction. Overall, *non-thermal* plasmonic-participation mechanism has partial role in selectivity enhancement which is further compensated by the pCOL's electronic effect.

Supplementary Figure 47. Plausible mechanism for Semihydrogenation of alkyne. Schematic representations of the semihydrogenation mechanism involved on the (a) pCOL-Pd/AuNC and (b) Pd/AuNC.

References

1. Rao, V. G., Aslam, U. & Linic, S. Chemical Requirement for Extracting Energetic Charge Carriers from Plasmonic Metal Nanoparticles to Perform Electron-Transfer Reactions. *J. Am. Chem. Soc.* **141**, 643–647 (2019).
2. Aslam, U., Chavez, S. & Linic, S. Controlling energy flow in multimetallic nanostructures for plasmonic catalysis. *Nat. Nanotechnol.* **12**, 1000–1005 (2017).
3. Kale, M. J. & Christopher, P. Plasmons at the interface. *Science* **349**, 587–588 (2015).
4. Li, K. *et al.* Balancing Near-Field Enhancement, Absorption, and Scattering for Effective Antenna–Reactor Plasmonic Photocatalysis. *Nano Lett.* **17**, 3710–3717 (2017).
5. Lou, Z., Fujitsuka, M. & Majima, T. Pt–Au Triangular Nanoprisms with Strong Dipole Plasmon Resonance for Hydrogen Generation Studied by Single-Particle Spectroscopy. *ACS Nano* **10**, 6299–6305 (2016).
6. Keblinski, P., Cahill, D. G., Bodapati, A., Sullivan, C. R. & Taton, T. A. Limits of localized heating by electromagnetically excited nanoparticles. *J. Appl. Phys.* **100**, 054305 (2006).
7. Pozzi, E. A. *et al.* Evaluating Single-Molecule Stokes and Anti-Stokes SERS for Nanoscale Thermometry. *J. Phys. Chem. C* **119**, 21116–21124 (2015).
8. Wang, F. *et al.* Plasmonic Harvesting of Light Energy for Suzuki Coupling Reactions. *J. Am. Chem. Soc.* **135**, 5588–5601 (2013).
9. Zhou, L. *et al.* Quantifying hot carrier and thermal contributions in plasmonic photocatalysis. *Science* **362**, 69–72 (2018).
10. Kazuma, E. & Kim, Y. Mechanistic Studies of Plasmon Chemistry on Metal Catalysts. *Angew. Chem. Int. Ed.* **58**, 4800–4808 (2019).
11. Huang, H. *et al.* Unraveling Surface Plasmon Decay in Core–Shell Nanostructures toward Broadband Light-Driven Catalytic Organic Synthesis. *J. Am. Chem. Soc.* **138**, 6822–6828 (2016).

12. Saikin, S. K., Olivares-Amaya, R., Rappoport, D., Stopa, M. & Aspuru-Guzik, A. On the chemical bonding effects in the Raman response: Benzenethiol adsorbed on silver clusters. *Phys. Chem. Chem. Phys.* **11**, 9401–9411 (2009).
13. Pérez-Jiménez, A. I., Lyu, D., Lu, Z., Liu, G. & Ren, B. Surface-enhanced Raman spectroscopy: benefits, trade-offs and future developments. *Chem. Sci.* **11**, 4563–4577 (2020).
14. Joplin, A. *et al.* Correlated Absorption and Scattering Spectroscopy of Individual Platinum-Decorated Gold Nanorods Reveals Strong Excitation Enhancement in the Nonplasmonic Metal. *ACS Nano* **11**, 12346–12357 (2017).
15. Quiroz, J. *et al.* Controlling Reaction Selectivity over Hybrid Plasmonic Nanocatalysts. *Nano Lett.* **18**, 7289–7297 (2018).
16. Peiris, E. *et al.* Plasmonic Switching of the Reaction Pathway: Visible-Light Irradiation Varies the Reactant Concentration at the Solid–Solution Interface of a Gold–Cobalt Catalyst. *Angew. Chem. Int. Ed.* **58**, 12032–12036 (2019).
17. Peiris, E. *et al.* Controlling Selectivity in Plasmonic Catalysis: Switching Reaction Pathway from Hydrogenation to Homocoupling Under Visible-Light Irradiation. *Angew. Chem. Int. Ed.* **62**, e202216398 (2023).
18. Swearer, D. F. *et al.* Heterometallic antenna–reactor complexes for photocatalysis. *Proc. Natl. Acad. Sci. U.S.A.* **113**, 8916–8920 (2016).

REVIEWER COMMENTS

Reviewer #1 (Remarks to the Author):

The authors revised the MS perfectly well and answered most of my queries. I think the revised MS deserves publication in Nature Communications.

Reviewer #2 (Remarks to the Author):

The manuscript is now ready for acceptance. This is a very nice work. Congrats!

Reviewer #3 (Remarks to the Author):

I believe this manuscript has done a good job addressing all of the reviewer comments from the first round. The introduction is more compelling, and there are a lot more control experiments that support their conclusions. However, I have one comment that I believe should be addressed before publication in Nature Communications.

The schematic in Figure 1 and Figure S47 shows charge carriers being generated in the Au core and transferring to the Pd shell. It has been argued (ACS Energy Letters, 3, 7, 1590-1596, 2018) and experimentally shown (ACS Nano, 4, 14, 5061-5074, 2020) that adding a transition metal to the surface of the plasmonic metal leads to the coupling of s electrons in the plasmonic metal to d electrons in the transition metal, resulting in the direct localization/excitation of charge carriers in the transition metal shell. This conclusion is also supported by the authors transient absorption data, that show the lifetimes for the carriers are shorter when the Au systems are modified with Pd. Additionally, the linewidth of the UV vis spectra for the Pd modified systems are larger than the systems without Pd, again suggesting shorter carrier lifetimes with the Pd shell present. The authors should address this in their manuscript/figures.

Response to the Reviewer's comment:

Reviewer #3 (Remarks to the Author):

I believe this manuscript has done a good job addressing all of the reviewer comments from the first round. The introduction is more compelling, and there are a lot more control experiments that support their conclusions. However, I have one comment that I believe should be addressed before publication in Nature Communications.

The schematic in Figure 1 and Figure S47 shows charge carriers being generated in the Au core and transferring to the Pd shell. It has been argued (ACS Energy Letters, 3, 7, 1590-1596, 2018) and experimentally shown (ACS Nano, 4, 14, 5061-5074, 2020) that adding a transition metal to the surface of the plasmonic metal leads to the coupling of s electrons in the plasmonic metal to d electrons in the transition metal, resulting in the direct localization/excitation of charge carriers in the transition metal shell. This conclusion is also supported by the authors transient absorption data, that show the lifetimes for the carriers are shorter when the Au systems are modified with Pd. Additionally, the linewidth of the UV vis spectra for the Pd modified systems are larger than the systems without Pd, again suggesting shorter carrier lifetimes with the Pd shell present. The authors should address this in their manuscript/figures.

Response: As reviewer has suggested, we have modified the Figure 1 in the manuscript and Supplementary Fig. 47. We have also added the relevant text (Page 8, line 22-25) and references (Ref 44,45) in the revised manuscript.

REVIEWERS' COMMENTS

Reviewer #3 (Remarks to the Author):

The reviewers have significantly improved the manuscript during the review process. It is ready to be published in Nature Communications. Congrats to the authors.